# On fluctuating air-sea-interaction in local models: linear theory

Achim Wirth[1]

[1]Univ. Grenoble Alpes, CNRS, Grenoble INP, LEGI, 38000 Grenoble, France

**Correspondence:** achim.wirth@legi.cnrs.fr

**Abstract.** The dynamics of three local linear models of air sea-interation commonly employed in climate or ocean simulations is compared. The models differ by whether or not the ocean velocity is included in the shear calculation applied to the ocean and the atmosphere. Analytic calculations for the models with deteministic and random forcing (white and colored) are presented. The short term behavior is similar in all models, which only small quantitative differences, while the longterm behavior differs qualitatively between the models. The fluctuation-dissipation-relation, which connects the fast excitation to the slow dissipation, is establised for all models with random forcing. The fluctuation-dissipation-theorem, which compares the response to an external forcing to internal fluctuations is established for a white-noise forcing and a colored forcing when the phase space is augmented by the forcing variable. Using results from numerical integrations of stochastic differential equations shows that the fluctuation-theorem, which compares the probability of positive to negative fluxes of the same magnitude, averaged over time-intervals of varying length, holds for the energy gained by the ocean from the atmosphere.

## 1 Introduction

The exchange of momentum, heat, water and chemical fluxes at the atmosphere-ocean interface is key to understanding the dynamics of the atmosphere, the ocean and the climate, as well as their response to changes in the forcing of the climate system (Stocker et al. (2013), Csanady (2001)). In the present paper the exchange of momentum, only, is considered. The exchange is usually parameterized by local models (so called bulk formulas), which imitate the action of the non-explicitly resolved dynamics on the explicitely resolved dynamics in atmosphere, ocean and climate models. In the present paper the behavior of the most widely used local models subject to deterministic and random forcing are discussed. Only the local exchange between the atmosphere and the ocean is considered, neglecting the horizontal interaction within the atmosphere and the ocean. Mathematically speaking the models are zero-dimensional one-component (0D1C) (see section 2). The linear models used here allow for analytic solutions. When within a hierarchy of models a systematic liaison of the more involved models to models that allow for an analytic solution can be established, the scientific understanding of the process studied is increased (Wirth (2010)).

The conspicuous feature of the atmosphere ocean system is the strong difference in mass (and also heat capacity, $CO_2$ absorption) of the two media, leading to strong difference in the characteristic time scales for inertia (and also heat, $CO_2$ storage). In this respect there is a strong analogy to Brownian motion, with light and fast molecules colliding with heavy and slow Brownian particles. The Fluctuation-Dissipation-Relation (FDR) developed by Einstein (1906) represents the framework to describe such motion. He noted that a Brownian particle in a fluid is subject to two processes, a macroscopic friction and microscopic fluctuations, which are related as they are both due the surrounding fluid (see Einstein (1906, 1956), Perrin (2014)). The FDR describes the relation between the two processes (Barrat and Hansen (2003)). The FDR is applied to a large variety of linear and non-linear problems in the field of non-equilibrium statistical mechanics, also when the “Brownian particle” is some “slow” property of a system. In air-sea interaction the friction at the interface dissipates energy and introduces fluctuations in both media. Also in this case, dissipation and fluctuations are due to the same process and a relation between the two has to exist. This relation, the FDR, is established for the three models, subject to white and colored random forcing, in the present work.

The major difference between Brownian motion and air-sea interaction is that the former system is conservative while the latter is dissipative and forced from the exterior. Mathematically speaking: in the former, the dynamics conserves the phase space volume, while in the latter it contracts and the dynamics takes place on a strange attractor of vanishing phase space volume. A key feature of Brownian motion is the equipartition of energy between a Brownian particle and a molecule (Einstein (1906)), in the case of air-sea interaction equipartition does not hold. Although there are fundamental differences between conservative and dissipative-and-forced dynamics, many of the mathematical concepts developed can be extended from the former to the latter (Marconi et al. (2008)). In a previous publication (Wirth (2018)) the FDR was derived for one model of the atmosphere-ocean system, where the forcing on the atmosphere was white-in-time. This FDR was then compared to a two dimensional numerical simulation of air-sea interaction, giving rise to turbulent dynamics. The two-dimensional numerical simulation was forced by maintaining one Fourier-mode of the dynamics at a fixed value. The application of the FDR to 2D numerical simulations of turbulent dynamics succeeded for the ocean dynamics but failed for the atmosphere, as the former evolves on a time scale much slower than the forcing, while the time scale of the latter is equal to the forcing, which acts by restoring to a constant velocity profile. That is, there was no separation between the time scale of the forcing and the atmospheric dynamics in the 2D numerical simulations. The white-in-time FDR model, while successful for describing the ocean response, was therefore inappropriate to describe the atmospheric dynamics.

In the context of a purely 2D dynamics the energy dissipation within the atmospheric and the oceanic layer, due to horizontal friction processes, decreases with an increasing Reynolds number, due to the inverse cascade of energy in two-dimensional turbulence (Boffetta and Ecke (2012)). In the real ocean internal energy dissipation depends on a variety of processes as frontal dynamics, tidal mixing, stratification and bottom friction (Ferrari and Wunsch (2009)).

In this work three different mathematical models for parameterizing air-sea interaction which are all used in numerical models of climate dynamics are discussed. They differ by to what extend the ocean velocity is considered in the calculation of the shear force at the air-sea interface. In the first, the ocean velocity is ignored and shear is calculated based on the atmospheric velocity only. Historically this was done in all climate models and is justified by the fact that usually atmospheric

winds have higher speeds than ocean currents. In the second, the ocean velocity is considered when the shear-force applied to the ocean is calculated, but not to the atmosphere. These two models are called "one-way" as the ocean dynamic does not act on the atmosphere, they are used, for example, whenever the atmospheric forcing is known prior to the integration of the ocean model, when an ocean only simulation is performed. Only the third model is mechanically consistent as the shear force, applied to the ocean and the atmosphere, is calculated based on the difference between the atmospheric and the oceanic velocity vectors and respects Newton's laws. This model is called "two-way". In the present work which is mostly focused on analytic theory, the calculation of the shear is performed using a linear (Rayleigh) law.

We demonstrated in recent work using a two-dimensional model, that the third parameterization together with a quadratic drag law can lead to a generation of instability (Moulin and Wirth (2014)) and new dynamical behavior (Moulin and Wirth (2016)), that is a co-organization between the atmospheric and oceanic variables that resembles a glass-transition in condensed matter physics. It is shown in Wirth (2018), by solving the Fokker-Planck equation, that the second order moments of the non-linear model can be reproduced by a linear model using an eddy-friction approach with an eddy coefficient that is obtained analytically.

Here, only linear models are considered, as the focus is on the analytic theory (where possible). The analytic solution of a linear model gives the dependence on all parameters in a nonlinear model parameter dependence has to be numerically evaluated for each parameter. Furthermore, in the linear models, solutions with different forcing can be simply added up, in their non-linear counterpart this is no-longer true. The prolongation to non-linear models and their numerical solutions, will be discussed elsewhere. It is furthermore important to note that the major differences between the three models already emerge in their linear versions.

The present work compares the three different parameterizations of air-sea transfer of momentum discussed above with a linear drag law having a constant or periodic (deterministic) forcing or a white or colored random-forcing. This leads to $3 \cdot (2+2) = 12$ different local configurations which are discussed here.

The models are introduced in section 2 and the solutions with different forcing are discussed in section 3. The resulting FDRs for linear models with stochastic forcing are given in section 4. The results of the FDR are used to establish in section 5 the work done on the air-sea system, the energy fluxes and energy dissipation in the different models. A special emphasis is directed towards the consistency of the different models and their differences, quantitative and qualitative, for short- and long-term integrations.

The Fluctuation-Dissipation-Theorem (FDT) (see i.e. Kubo (1966), Barrat and Hansen (2003), Marconi et al. (2008)) is discussed in section 6. It considers the relation of the internal fluctuations of a system to the response of an external force. When it holds the average response of the system to an external force can be obtained by observing the average relaxation of spontaneous fluctuations. The FDT is today used in a large variety of statistical and dynamical systems as i.e. climate dynamics. If it holds for our climate system it allows for determining the response to perturbations, antropogenic or others, by studying its natural variability (see *e.g.* Cooper and Haynes (2011)). In the case of the linear models discussed here the FDT can be established analytically using matrix calculus. Studying the FDT in air-sea interaction is also interesting from a conceptual

Geoscientific Model Development Discussions Open Access EGU

point of view as air-sea interaction posses a dynamics on dissimilar and interacting time scales, a property found in many natural applications and processes of the climate system.

The Fluctuation Theorems (FTs) (Gallavotti and Cohen (1995a), Gallavotti and Cohen (1995b), Ciliberto et al. (2004)) for the energy exchange between the atmosphere and the ocean are numerically evaluated in section 7. When the atmosphere is force by a white or colored noise, the velocity fluctuations in the atmosphere have a larger amplitude than in the ocean and on average the atmosphere does work on the ocean and the ocean receives energy form the atmosphere. Instantaneous energy fluxes can however also go in the opposite direction. The probability of positive versus negative fluxes, averaged over finite time intervals of varying length, are subject of the FT. The results are discussed in section 8 and conclusions are presented in section 9

The analytic calculations are found in the vast appendixes, which form the major part of the publication. They present a register for calculations concerning local models of air-sea interaction. These calculations are important as they expose the strength and weaknesses of and the differences between the models.

## 2 Local models

The turbulent friction at the atmosphere-ocean interface is commonly modeled by a quadratic friction law, where the friction force is a constant times the product of the shear speed and the shear velocity (see e.g. Stull (2012)). The linear version with a constant eddy-coefficient allows for analytic solutions. It is also sometimes used in numerical simulations of the climate dynamics. The friction coefficient represents an average (in time and space) mimicking the real friction process.

The mathematical models discussed here are non-dimensionalized. The mass of the atmosphere per unit area is set to unity. The mass of the ocean per unit area is $m$ times the mass of the atmosphere, the total mass per unit area is $M = m + 1$. When the interaction of the atmospheric and oceanic mixed-layers are considered $m \approx O(10^2)$. Three linear models commonly used are discussed. These models give rise to different configurations which differ by the forcing. In the following equations $S$ is the inverse of the friction time in the ocean. When a linear model is used ($S$=const) both horizontal directions are un-coupled and the problem can be considered independently for each direction and reduces to a one dimensional problem. Scalar variables are therefore employed. Newton's third law sets the inverse friction time for the atmosphere to $Sm$. The forcing of the system is denoted $\tilde{F}$. In the first model, L1, the ocean velocity is not considered in the calculation of the shear force at the interface, neither in the dynamics of the atmosphere nor the ocean:

$$\partial_t u_a^{L1} = -Sm u_a^{L1} + \tilde{F} \quad (1)$$
$$\partial_t u_o^{L1} = S \; u_a^{L1}. \quad (2)$$

Note that this model is inconsistent as the ocean is accelerated in the direction of the atmospheric velocity even when the ocean is moving in the same direction with a higher speed. The model is justified by the observation that atmospheric wind-speeds are often much larger than ocean currents and when the difference of the two is considered the latter can be neglected to first order in $u_o/u_a$. The L1 represents the classical approach to implementing air-sea interaction.

In the second model, L2, the ocean velocity is considered in the calculation of the shear force at the interface when the ocean dynamics is considered, but not for the atmospheric velocity:

$$\partial_t u_a^{L2} = -Smu_a^{L2} + \tilde{F} \tag{3}$$

$$\partial_t u_o^{L2} = S \, (u_a^{L2} - u_o^{L2}). \tag{4}$$

Note that this model is inconsistent as interfacial friction does not conserve total (atmosphere plus ocean) momentum. This model neglects the action of ocean currents on the atmospheric dynamics, it is used, for example, in ocean modeling when ocean models are forced by winds which are predefined, available previous to the ocean simulation. The L1 and L2 models are called “one-way”.

In L3 the ocean velocity is considered in the calculation of the shear force at the interface, when the atmosphere and ocean dynamics is considered:

$$\partial_t u_a^{L3} = -Sm(u_a^{L3} - u_o^{L3}) + \tilde{F} \tag{5}$$

$$\partial_t u_o^{L3} = S \, (u_a^{L3} - u_o^{L3}). \tag{6}$$

This model obeys Newton’s laws. Including the ocean velocity on the r.h.s. of eqs. (4) and (6) damps the ocean velocity and is recently referred to as the “eddy killing” term. It is found to have a considerable effect on the ocean dynamics (see i.e. Renault et al. (2017)).

For each of the linear models four different kinds of forcing are distinguihed. The first is constant forcing starting at a time $t = t_0$. The solutions are discussed in the next section and given in the appendix B1. These configurations are denoted LxK ($x = 1,2,3$) for the three different models mentioned above. The second is periodic forcing $\tilde{F} = \cos(\omega t)$. These models are denoted LxP ($x = 1,2,3$). The solutions for the atmospheric and the oceanic velocities as well as the second-order moments, obtained by averaging over one period of $\tau = 2\pi/\omega$ are discussed in the next section and given in the appendix B5.

White-in-time random forcing $F$ is also considered. The parameter $R$ measures the strength of the delta-correlated fluctuating force, it is:

$$R = \int_0^\infty \langle F(0)F(t')\rangle_\Omega dt'. \tag{7}$$

In configurations LxW ($x = 1,2,3$) the atmosphere is directly forced by a the white noise, $\tilde{F} = F$.

In the fourth series of configurations, LxC ($x = 1,2,3$), the same models are forced using a colored noise, which is itself a solution of a Langevin equation:

$$\partial_t \tilde{F} = -\mu \tilde{F} + F \tag{8}$$

It is sometimes stated that without a clear-cut separation between the relaxation time-scale and the noise correlation-time the process is non-Markovian. This problem is avoided here by using a colored noise ($\tilde{F}$ has the finite correlation-time $\mu^{-1}$), which is itself generated by a white-noise through a linear Langevin equation. Indeed, when $F$ is white in time, the variable $\tilde{F}$

is an Ornstein-Uhlenbeck process. When the atmosphere-ocean system is forced by $\tilde{F}$ and the three variable system $(\tilde{F}, u_a, u_o)$ is considered, the system is a Markov-process, while the problem is non-Markovain in the two variable system $(u_a, u_o)$. Augmenting the phase space dimension to render a non-Markovian process Markovian is a standard procedure.

## 3 Solutions of local models

In the local linear models all solutions are analytic, for all types of forcing considered. These models are a firm testing ground for all theories and statements on air-sea interaction.

First, the unforced evolution of an initial state in the three models is compared. For the consistent model L3 eq. (A26) shows, that the total inertia in the system $(u_a + m u_o)$ is conserved and the shear between the atmosphere and the ocean $|u_a - u_o|$ decays with a characteristic time-scale of $(SM)^{-1}$. For the L1 model eq. (A12) shows that the total inertia is conserved and every perturbation in the atmosphere decays with a characteristic time scale of $(Sm)^{-1}$ and adds to the the ocean $m^{-1}$ times the initial atmospheric perturbation at the same characteristic time scale. Ultimately all the inertia is in the ocean which has no influence on the atmosphere. In the L2 model perturbations in the atmosphere and the ocean decay and a spurious slow time-scale $S^{-1}$ appears in the ocean dynamics, as can be seen from eq. (A19) . Replacing the L1 model by the L2 model is not always an improvement as it leads to a decay of all motion and introduces an artificial time-scale.

Second, the solutions of the different models, subject to the same forcing, are compared. Only the atmosphere is subject to an external forcing. Two extreme cases can be distinguished. The first is the short term response and the second is the long term evolution. To consider the first question only the LxK configurations (see appendix B1), in which a constant forcing is turned on at $t = t_0$, have to be considered. Every forcing can be decomposed into a sum (integral) over (possibly infinitesimal) step functions. As the model is linear its dynamics is a sum (integral) over a (finite or infinite number) of solutions with a single (possibly infinitesimal) step. From the Taylor-series expansion of the LxK configurations in appendices B1 it emerges that the short time response of the L1 and the L2 model is similar to the L3 model as the first terms agree, for the atmosphere and the ocean and the two consecutive terms have coefficients which differ by at most a factor of the order of $m/M \approx 1$.

The long-term behaviors with constant forcing of the atmosphere are however completely different (see appendix B1). For the L3 model, both the atmospheric and oceanic velocity are unbounded and increase at the same rate. For the L1 model, the atmospheric velocity is bounded while the oceanic velocity is unbounded and increases at a rate that is $M/m \approx 1$ higher as compared to the L3 model. For the L2 model, both the atmospheric velocity and oceanic velocity are bounded. So differences are not only quantitative but also qualitative and the L1 performs better in a coupled simulation, when the ocean dynamics is considered. This is important to note, as it is the ocean dynamics community that favors a passage from the L1 to the L2 parameterization.

When the forcing applied to the atmosphere is periodic ($F_a = \cos(\kappa t)$, see appendix B5) all solutions are periodic. The ratio of the square amplitude of the ocean and the atmospheric velocities and their normalized correlations are:

$$\Xi = \frac{\langle u_o^2 \rangle_\tau}{\langle u_a^2 \rangle_\tau} \tag{9}$$

$$\Theta = \frac{\langle u_a u_o \rangle_\tau}{\sqrt{\langle u_a^2 \rangle_\tau \langle u_o^2 \rangle_\tau}}, \tag{10}$$

where averages are taken over one period $\tau = 2\pi/\kappa$. In the L3P model $\Xi = S^2/(S^2+\kappa^2)$ is always smaller than one and oceanic velocities approach the atmospheric velocities, when the characteristic forcing time scale increases. The normalized correlation $\Theta = S/\sqrt{S^2+\kappa^2} = \sqrt{\Xi}$ shows that the slower the forcing the higher is the correlation between the atmospheric and oceanic velocities. For the L1P model, $\Xi = S^2/\kappa^2$ which approaches the consistent L3P model for a high frequency forcing, only. Values larger than unity are however non-physical and so a forcing in which the oceanic forcing-time is larger than the oceanic friction-time can not be considered with this model. This is worrisome as a forcing of the atmospheric system contains components of arbitrary long time-scales. Furthermore, $\Theta = 0$, which means that the phase shift between the atmosphere and ocean is $\pi/2$. For the L2P model $\Xi$ and $\Theta$ are identical to the L3P model and it is clearly a better choice than the L1P model.

Some of the models with random forcing have a dynamics which is not statistically stationary and time averages depend on the length of the averaging interval. Time averages are therefore replaced by ensemble averages, taken over an ensemble ($\omega \in \Omega$) of realizations of forcing functions $F_\omega$, they are denoted by $\langle . \rangle_\Omega$, where $\omega$ is one realization of the ensemble space $\Omega$. The dynamics starts from rest at $t_0 = 0$, for convenience. When the forcing is Gaussian in the LxW and LxC configurations the probability density functions (pdfs) of the variables $\tilde{F}, u_a, u_o$ are centered Gaussians, first order moments vanish and the pdfs are determined by their second-order moments. Using stochastic calculus (see appendix B9 and Wirth (2018)) the second-order moments are obtained analytically.

The differences in the results of the models are, again, not only quantitative but qualitative. The total momentum, atmosphere plus ocean, performs a random-walk (see appendix B9B10) in the L3W and L3C models, as it is not subject to damping. Superposed to this motion is the shear mode, which performs an Ornstein-Uhlenbeck process (see appendix B9B10). The L1 model has an atmospheric mode which performs an Ornstein-Uhlenbeck process and the oceanic mode that performs a random walk which is forced by the atmospheric dynamics on which it does not retro-act. In the L2 model a damping is added to the ocean mode as compared to the L1 model which leads to an Ornstein-Uhlenbeck process also in the ocean. In the L3 model there is a constant and equal growth-rate of all second-order moments (see appendix B11). In the L1 model the linear growth is only present in the ocean with a growth rate which is by a factor $(M/m)^2 \approx 1$ higher than for the L3 model. In the L2 model all second order moments are bounded. All the results from analytical calculations are given in appendix B11 and tables 1 and 2. The differences between the models with white and colored noise (LxW and LxC) are only quantitative and tables 1 and 2 show that for short correlations times ($\mu \ll SM$) the dynamics of the colored-noise cases converges to the corresponding white-noise cases.

It is important to note that some of these models do not lead to a (statistically) stationary state, but their ensemble averages evolve in time. All the processes are, however, of stationary increment, that is, the time increments of random variables

$(u_a, u_o, F)$ and linear combinations thereof are stationary ($\langle x(t) - x(t+\Delta t)\rangle_\Omega$ depends on $\Delta t$ but not on $t$). The dynamics is a sum of Ornstein-Uhlenbeck processes and random walks, which are all of statistically stationary increment.

## 4 Fluctuation Dissipation Relation

At the interface the ocean (eqs. (2), (4) and (6)) is subject to forcing by the atmosphere, given by $Su_a$, and (in the L2 and L3 models) dissipation, given by $-Su_o$. They both are due to the same process and must be therefore related. In analogy to Brownian motion I call this relation, when expressed by the second-order moments, the oceanic Fluctuation Dissipation Relation (FDR). The atmosphere (eq. (1), (3) and (5)) is subject to outside forcing (given by $F$), dissipation given by $-Smu_a$ and, in the L3 model, to fluctuations by the ocean, given by $Smu_o$. The atmospheric FDR consist in the relation of the three processes in the equation of the second-order moments. Furthermore, the total momentum, atmosphere plus ocean, is subject to external forcing but not to internal dissipation in the case of the L1 and L3 model. In the L2 model, the total momentum is influenced by the forcing and the ocean velocity. The latter is nonphysical.

As an example the FDR in the configuration L3W is considered after the initial spin up, that is for $t \ll (SM)^{-1}$. The FDR is obtained by multiplying eq. (5) by $u_a$, eq. (6) by $u_o$ and ensemble averaging. The second-order moments are given by the correlation matrix (B41), which leads to:

$$\frac{1}{2}\partial_t \langle u_a^2\rangle_\Omega = \frac{SmR}{M^2}\left(\underbrace{2t + \frac{m-2}{SM}}_{\text{fluctuation}}\underbrace{-2t - \frac{m^2+4m}{SM}}_{\text{dissipation}}\right) + \underbrace{R}_{\text{forcing}} = \frac{R}{M^2} \tag{11}$$

$$\frac{1}{2}\partial_t \langle u_o^2\rangle_\Omega = \frac{SR}{M^2}\left(\underbrace{2t + \frac{m-2}{SM}}_{\text{fluctuation}}\underbrace{-2t + \frac{3}{SM}}_{\text{dissipation}}\right) = \frac{R}{M^2} \tag{12}$$

$$\frac{1}{2}\partial_t \langle u_a^2 + mu_o^2\rangle_\Omega = \frac{SmR}{M^2}\left(\underbrace{-\frac{M}{S}}_{\text{dissipation}}\right) + \underbrace{R}_{\text{forcing}} = \frac{R}{M}. \tag{13}$$

For the atmosphere (eq. (11)) the fluctuation terms are due to the atmosphere-ocean correlation, on average they drive the atmospheric dynamics, as the ocean dynamics reduces the friction, on average. The dissipation terms are due to the atmospheric auto-correlation. The forcing term drives the atmospheric dynamics and the whole system. The sum of the fluctuating, the dissipation and the force leads to a constant-in-time increase of the square velocity in the atmosphere.

Concerning the ocean (eq. (12)) the fluctuation terms originate from the atmosphere-ocean correlation, they are driving the ocean dynamics, on average. The oceanic auto-correlation leads to the dissipation terms. The first and third term, due to the total-inertia mode in the atmosphere and the ocean cancel, as the total inertia mode performs a random walk (Wiener-process) with no shear associated to it. The second and forth term due to the shear-mode in the atmosphere and the ocean, respectively, lead to a statistically stationary dynamics, an Ornstein-Uhlenbeck process. The sum the fluctuating force and the dissipation leads to a constant-in-time increase of the square velocity in the ocean which is equal to the increase in the atmosphere. The

atmospheric fluctuations are the only forcing acting on the ocean The total energy is forced by the exterior and dissipated due to the internal shear (eq. (13)).

Equations (11) and (12) show the equilibrium between the fluctuation-dissipation, the forcing and the energy growth. This is a double fluctuation-dissipation relation: the dissipation and the fluctuation are related, firstly, by the equal growth rate of their squares and secondly by the difference between them. Brownian motion leads to an equipartition of energy between molecules and Brownian particles, which, in air-sea interaction is substituted by the equal growth rate of the square velocities of the atmosphere, the ocean and their covariance.

When observing the second-order moments the parameters in the linear model are given by:

$$S = \frac{\partial_t \langle u_o^2 \rangle_\Omega}{2\langle u_a u_o - u_o^2 \rangle_\Omega} \tag{14}$$

$$M = \frac{\langle u_a^2 - u_o^2 \rangle_\Omega}{\langle u_a u_o - u_o^2 \rangle_\Omega}. \tag{15}$$

It is straightforward to determine the FDR for LW1 and LW2 using results from the appendix B11. In the L1 and L2 model the fluctuation is neglected in the atmosphere and only the forcing term and the dissipation term are present. In the L1 model the dissipation is neglected in the ocean and only the fluctuation is present and the ocean performs a random-walk.

A FDR can be established for the configurations with a forcing by a colored noise (LxC), in the same way (see appendix B11). They show qualitatively the same behavior than the corresponding configurations forced by a white noise.

It is essential to note that in the linear models discussed in this subsection the forcing can be a linear combination of different forcing proposed with different periods and correlation times. The second-order moments are the sum of the individual second-order moments, that is cross-correlations of variables with a different forcing vanish.

When the forcing is a combination of a random and a periodic forcing it is important to note that the periodic part does not contribute to the (linear) growth-rate and that it also does not contribute to the difference in the correlation between the ocean-variance and the ocean-atmosphere correlation, both facts are related. The periodic part is however important when it comes to evaluating the difference between the atmosphere- and ocean-variance. This is a possible explanation, why the estimation of the friction parameter was successful in Wirth (2018) but not the mass-ratio between the atmosphere and the ocean. The latter compares the atmosphere- and ocean-variance, while the former is based on the difference of the ocean-variance and the ocean-atmosphere correlation, only.

## 5 Energetics

The fluxes of kinetic energy in the system, are detailed in fig. 1. The forcing injects energy in the atmosphere, part of it leads to an increase of the energy of the atmosphere $P_a$ and part of it is transferred to the atmosphere-ocean interface $P_{a\to i}$. At the interface energy is transferred between the atmosphere and the ocean and also dissipated $P_{\text{dissip}}$. The energy flux to the ocean $P_{i\to o}$ leads to an energy increase $P_o$. The dynamics in the system does not, for all models, converge to a stationary state, in which $P_a = P_o = 0$.

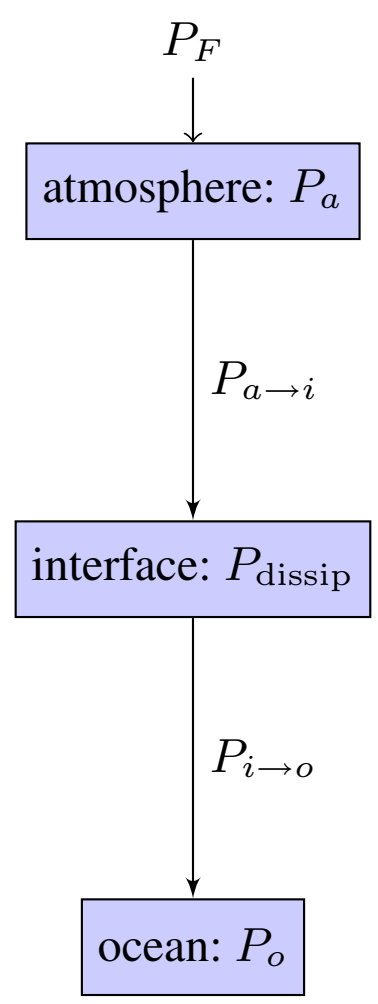

**Figure 1.** Schematic of energy fluxes in the atmosphere ocean system

When the forcing is periodic averages over one period are taken and when the forcing is stochastic ensemble averages are performed. For convenience the same symbol $\langle . \rangle$ denotes the averages, when the forcing is periodic $\langle . \rangle = \langle . \rangle_\tau$ and when the forcing is stochastic $\langle . \rangle = \langle . \rangle_\Omega$. The power injected in the system is $P_F = \langle u_a \tilde{F} \rangle$. The increase of energy in the atmosphere $P_a = \frac{1}{2}\partial_t \langle u_a^2 \rangle$ and the ocean $P_o = \frac{m}{2}\partial_t \langle u_o^2 \rangle$ are obtained by first multiplying eqs. (1, 3, 5) by $u_a$ and eqs. (2, 4, 6) by $u_o$ and then averaging. In the consistent model L3, the auto-correlation of the atmosphere and the ocean dissipate the energy in the atmosphere and the ocean, respectively. In the L1 model the dissipation in the ocean is omitted. By looking at the equations the important role of the correlation between the atmospheric and oceanic velocity (the fluctuation term) $\langle u_a u_o \rangle$ becomes clear. On average it is non-negative in all models and configurations. In the consistent model L3 it has two effects it reduces the energy dissipation in the atmosphere and injects the energy in the ocean, both are equal, not only in magnitude but also in sign $(+Sm\langle u_a u_o \rangle)$. In the L1 and L2 models the reduction of the energy dissipation in the atmosphere, due to ocean velocities, is omitted. Therefore, the L1 model suffers from an increased energy dissipation in the atmosphere and a reduced energy dissipation in the ocean, while in the L2 model only the first is present.

In all the models the fluxes are related by:

$$P_{a\to i} = P_F - P_a \tag{16}$$

$$P_{i\to o} = P_o \tag{17}$$

$$P_{\mathrm{dissip}} = P_{a\to i} - P_{i\to o} = P_F - P_a - P_o. \tag{18}$$

If the fluxes are time independent, or long-time averages are taken, they are all non-negative for the L2 and L3 model (see also the discussion of the Fluctuation Theorem in section 7). In the L1 model $P_{\mathrm{dissip}} < 0$ when $u_o u_a > u_a^2$, which is ofcourse unphysical.

| Exp. | $P_F$ | $P_a/P_F$ | $P_{a\to i}/P_F$ | $\frac{P_{\text{dissip}}}{P_F}$ | $P_o/P_F$ | $\frac{P_a+P_o}{P_F}$ | $P_a/P_{a\to i}$ | $\frac{P_a+P_o}{P_{\text{dissip}}}$ | $\eta$ |
|---|---|---|---|---|---|---|---|---|---|
| L1P | $\frac{Sm}{\kappa^2+(Sm)^2}$ | 0 | $\frac{Sm}{\kappa^2+(Sm)^2}$ | $\frac{Sm}{\kappa^2+(Sm)^2}$ | 0 | 0 | 0 | 0 | 0 |
| L2P | $\frac{Sm}{\kappa^2+(Sm)^2}$ | 0 | $\frac{Sm}{\kappa^2+(Sm)^2}$ | $\frac{Sm}{\kappa^2+(Sm)^2}$ | 0 | 0 | 0 | 0 | 0 |
| L3P | $\frac{Sm}{\kappa^2+(SM)^2}$ | 0 | $\frac{Sm}{\kappa^2+(SM)^2}$ | $\frac{Sm}{\kappa^2+(SM)^2}$ | 0 | 0 | 0 | 0 | 0 |
| L1W | R | 0 | 1 | $\frac{m^2-1}{m^2}$ | $\frac{1}{m^2}$ | $\frac{1}{m^2}$ | 0 | $\frac{1}{m^2-1}$ | $\frac{1}{m^2}$ |
| L2W | R | 0 | 1 | 1 | 0 | 0 | 0 | 0 | 0 |
| L3W | R | $\frac{1}{M^2}$ | $\frac{M^2-1}{M^2}$ | $\frac{m}{M}$ | $\frac{m}{M^2}$ | $\frac{1}{M}$ | $\frac{1}{M^2-1}$ | $\frac{1}{m}$ | $\frac{1}{M+1}$ |
| L1C | $\frac{R}{\mu(\mu+Sm)}$ | 0 | 1 | $\frac{\mu(m^2-1)-Sm}{\mu m^2}$ | $\frac{\mu+Sm}{\mu m^2}$ | $\frac{\mu+Sm}{\mu m^2}$ | 0 | $\frac{\mu+Sm}{\mu(m^2-1)-Sm}$ | $\frac{\mu-Sm}{\mu m^2}$ |
| L2C | $\frac{R}{\mu(\mu+SM)}$ | 0 | 1 | 1 | 0 | 0 | 0 | 0 | 0 |
| L3C | $\frac{R(\mu+S)}{\mu^2(\mu+SM)}$ | $\frac{\mu+SM}{M^2(\mu+S)}$ | $\frac{\mu(M^2-1)+SMm}{M^2(\mu+S)}$ | $\frac{\mu m}{M(\mu+S)}$ | $\frac{m(\mu+SM)}{M^2(\mu+S)}$ | $\frac{\mu+SM}{M(\mu+S)}$ | $\frac{\mu+SM}{\mu(M^2-1)+SMm}$ | $\frac{\mu+SM}{\mu m}$ | $\frac{\mu+SM}{\mu(M+1)+SM}$ |

**Table 1.** Energy fluxes for $t \gg (SM)^{-1}, \mu^{-1}$. The last column is the efficiency in the system as it compares the energy growth in the system to the energy injection. Note that for $\mu \gg SM$, LCx converges to LWx if $R \to R\mu^2$.

Using eqs. (16) - (18) allows to calculate the remaining energy fluxes. Note that in L3, $P_{\text{dissip}} = Sm\langle(u_a - u_o)^2\rangle$. Detailed results for all the energy fluxes are given in table 1.

The efficiency of the power transfer, is the power gained by the ocean at the interface divided by the power lost by the atmosphere at the interface:

$$\eta \quad = \quad \frac{P_{i\to o}}{P_{a\to i}} = 1 - \frac{P_{\text{dissip}}}{P_{a\to i}}. \tag{19}$$

The important question of the efficiency of the power transfer in the air-sea system, its dependence on the parameters and its representation in different models, has, to the best of my knowledge, never been addressed. Note that when no averaging is performed, the instantaneous fluxes in a single experiment is considered, $\eta = u_o/u_a$ in the L1 and L3 model, while $\eta = (1 - u_o/u_a)u_o/u_a$ for L2. When $\eta > 1$ the ocean is providing energy to the atmosphere. When the constant forcing is considered, the initial behavior of the efficiency is identical, to leading order, in the three model. The long-term behaviors differ: in L1 $\eta$ grows linearly to infinity, in L2 it converges to $\eta = 0$ and in L3 int converges to $\eta = 1$. A striking feature, shown in table 1, is that for the different models the efficiency is of different order in the mass ratio $m$, when random forcing, white or colored, is applied. So again, the differences are not only quantitative, expressed by different prefactors, but they are clearly qualitative.

For the L3 model, the only model that respect Newton's laws, all second order moments have the same constant growth rate and so the differences of these second order moments are constant in time, they are given in table 2.

In a perfect gas in equilibrium with atoms of different mass the kinetic energy of each atom, measured by the temperature, is equal on average and heat flows on average from the hotter to the colder substance (second law of thermodynamics). For the forced and dissipative air-sea interaction of the L2 and L3 models, the energetic influence of the interface on the ocean is: $Sm(u_a u_o - u_o^2)$ which shows that a necessary condition for the ocean to receive energy at the interface is: $u_a^2 > u_o^2$. This is also true when averages are taken $\langle u_a^2\rangle > \langle u_o^2\rangle$, a consequence of the Cauchy-Schwarz inequality. This is reflected in the results presented in table 2, as all entries of the third column, giving $\langle u_a^2 - u_o^2\rangle$ for the L2 and L3 model are positive. Due to

| Exp. | $\langle u_a^2 - u_a u_o\rangle$ | $\langle u_a u_o - u_o^2\rangle$ | $\langle u_a^2 - u_o^2\rangle$ | $\langle (u_a - u_o)^2\rangle$ |
|---|---|---|---|---|
| L1P | 1 | $-\frac{S^2}{\kappa^2}$ | $1-\frac{S^2}{\kappa^2}$ | $1-\frac{S^2}{\kappa^2}$ |
| L2P | $\frac{\kappa^2}{S^2+\kappa^2}$ | 0 | $\frac{\kappa^2}{S^2+\kappa^2}$ | $\frac{\kappa^2}{S^2+\kappa^2}$ |
| L3P | 1 | 0 | 1 | 1 |
| L1W | $\frac{m-1}{Sm^2}$ | t-dep. | t-dep. | t-dep. |
| L2W | $\frac{m}{SM^2}$ | 0 | $\frac{m}{SM^2}$ | $\frac{m}{SM^2}$ |
| L3W | $\frac{M+1}{SM^2}$ | $\frac{1}{SM^2}$ | $\frac{M+2}{SM^2}$ | $\frac{M}{SM^2}$ |
| L1C | $\frac{(m-1)\mu-Sm}{Sm^2(\mu+SM)}$ | t-dep. | t-dep. | t-dep. |
| L2C | $\frac{\mu^2}{SM(S+\mu)(Sm+\mu)}$ | 0 | $\frac{\mu^2}{SM(S+\mu)(Sm+\mu)}$ | $\frac{\mu^2}{SM(S+\mu)(Sm+\mu)}$ |
| L3C | $\frac{SM+(M+1)\mu}{SM^2(SM+\mu)}$ | $\frac{1}{SM^2}$ | $\frac{2SM+(M+2)\mu}{SM^2(SM+\mu)}$ | $\frac{M\mu}{SM^2(SM+\mu)}$ |

**Table 2.** Differences of second-order-moments of the velocity (normalized by $2(\kappa^2+(Sm)^2)$ for L1P and L2P, by $2(\kappa^2+(SM)^2)$ for L3P, by R for LxW and by $R/\mu^2$ for LxC). Note that for $\mu \gg SM$, LxC converges to LxW

the symmetry of the L3 model it is evident that a necessary condition for the atmosphere to receive energy at the interface is $u_o^2 > u_a^2$. It is important to note that a less energetic forced atmosphere can do work on the ocean (when $m > 1$) . In the L1 model the ocean receives energy whenever $u_a \cdot u_o > 0$.

## 6 Fluctuation Dissipation Theorem : Response Theory

The Fluctuation Dissipation Theorem (FDT) compares the response of the system subject to an external perturbation to the internal fluctuations of the system. This is related to the Onsager's principle which states that the system relaxes from a forced state to the unforced dynamics, in the same manner as if the forced state were due to an internal fluctuation of the system. The expressions FDT, Onsager's principle and also response theory are often interchanged in applications precise definitions are given in appendixes C1 and C2. Mathematically speaking: if $\mathbf{x}(t)$ is the state vector of the system the correlation matrix is $C(t,\Delta t) = \langle \mathbf{x}(t)\mathbf{x}^t(t+\Delta t)\rangle$ and the normalized correlation matrix is $C(t,\Delta t)C(t,0)^{-1}$. The average decay of a perturbation $\bar{\mathbf{x}}$ is given by the perturbation matrix, $\langle \mathbf{x}(t+\Delta t)\rangle = \chi(t,\Delta t)\bar{\mathbf{x}}$. The FDT holds if:

$$C(t,\Delta t)C(t,0)^{-1} = \chi(t,\Delta t). \tag{20}$$

The processes considered here are of stationary increment and the perturbation matrices are independent of the actual time $t$ and so are the normalized correlation matrices (see also appendices A1 - A3 and B11). The application of the FDT in the case of simple Langevin equations is discussed in appendix C1 for the white-noise forcing and in appendix C2, for forcing with a colored-noise.

The calculations concerning the application of the FDT for the models considered is given in appendix B9 eq. (B38), which show that the FDT applies in the models with the white-noise forcing and in the model with the colored-noise forcing, when the phase space is augmented by the variable $\tilde{F}$ representing the colored-noise forcing. Indeed, the FDT can be verified in these

models by explicitly calculating and comparing the matrices in eq. (20). The normalized correlation matrix with a white noise forcing, which is equal to the perturbation matrix, is given for the L1 model in eq. (A12), for the L2 model in eq. (A19) and for the L3 model in eq. (A26). The decay of an initial perturbation in the models was discussed in detail at the beginning of section 3. In the L1 model the ocean dynamics is undamped (see eq. (A12)) and performs a random-walk. In the L3 model the total momentum is undamped (see eq. (A26)) and performs a random-walk. The random walk, has the martingale property, that is the expectation for future values is equal to the actual value. This leads to an infinite memory in the process and infinite long correlations $\langle u_{total}(t)u_{total}(t+\Delta t)\rangle = \langle u_{total}(t)^2\rangle \ \forall \Delta t > 0$. Even in the case of the random walk, where the perturbation (forced or internal) does not decay, the FDT applies, as eq. (20) is verified.

For the colored-noise forcing the perturbation matrix in the augmented phase space is given for the L1C model in eq. (B53), for the L2C model in eq. (B64) and for the L3C model in eq. (B76). The failure of the FDT for a colored-noise forcing in a phase space consisting only of the velocities is due to the fact that the forcing has a finite correlation time and so the future forcing is correlated to the actual velocities and the correlations of the velocities is not the same as described by the perturbation matrix. The decay of a perturbation is therefore dependent on how the perturbation was reached, that is the system is not Markovian. This is shown in appendix C2 and discussed in detail by Balakrishnan (1979).

The FDT relies strongly on Gaussian statistics (see *e.g.* Cooper and Haynes (2011)) of the variables, which is assured in linear-models but the statistics in non-linear models is clearly non-Gaussian. As, to the best of my knowledge, no analytic solution exists for the non-linear versions of the air-sea interaction models the FDT has to be explored by numerical experiment. This will be done elsewhere.

## 7 Fluctuation Theorem

The average states and fluxes in the different models investigated as a function of their parameters are given in section 5. The probability density functions of the variables representing the atmospheric and ocean velocities are centered Gaussian variables and they are therefore completely described by their variance. Fluxes are products of Gaussian random-variables and are not Gaussian. The present section discusses instantaneous deviations from the average values, the fluctuations of the system and their persistence in time. To this end the Fluctuation Theorem (FT) (Gallavotti and Cohen (1995a), Gallavotti and Cohen (1995b), Ciliberto et al. (2004)) is discussed for the fluxes $P_{a\to i;\omega}$ and $P_{i\to o;\omega}$. The FT expresses properties of these quantities evaluated along fluctuating trajectories (indexed by $\omega \in \Omega$). The second law of thermodynamics states that heat always flows spontaneously from hotter to colder bodies. The FT specifies that this property is true on average but locally in-time-and-space counter fluxes are present. The relation of the probability of positive versus negative fluxes of a given magnitude and their persistence in time is subject of the FT.

The concepts of the FT are applied to a variety of problems and quantities and is also extended to deterministic dynamical systems. In the present work the analysis of Ciliberto et al. (2004) who tested the FT in two examples of turbulent flows in laboratory experiments is applied. Section 5 showed that on average the atmosphere gains energy by the random forcing and looses energy at the interface and the ocean gains energy at the interface. Also in the case of air-sea interaction instantaneous

fluxes can go in the opposite direction (Moulin and Wirth (2016)). The FT quantifies the asymmetry of the pdf of averages of the fluxes with respect to zero. It compares the probability of having a positive event to the probability of having a negative event of the same magnitude for averages of the fluxes over a time interval $\tau$. Do the symmetries implied by the Fluctuation Theorem (FT) apply to the momentum fluxes $P_{a\to i;\omega}$ and $P_{i\to o;\omega}$? The fluxes are quadratic quantities and their statistics is therefore not Gaussian. Recently a closed form of the probability density function $f(Z)$ of the product of two correlated Gaussian variables $Z = XY$ with vanishing means and variances $\sigma_x^2$ and $\sigma_y^2$ and correlation $\rho$ has been obtained (Nadarajah and Pogány (2016) and Gaunt (2018)) in terms of a modified Bessel function of the second kind of order zero $K_0(z) = \int_0^\infty \cos(z \sinh t)dt$:

$$f(z) = \frac{1}{\pi\sigma_x\sigma_y\sqrt{1-\rho^2}} \exp\left(\frac{\rho z}{\sigma_x\sigma_y(1-\rho^2)}\right) K_0\left(\frac{|z|}{\sigma_x\sigma_y(1-\rho^2)}\right). \tag{21}$$

The corresponding symmetry function is:

$$S_Z(z) = \ln\left(\frac{f(z)}{f(-z)}\right) = \beta z, \tag{22}$$

which is linear in the variable $z$ and the prefactor is $\beta = 2\rho/((1-\rho^2)\sigma_x\sigma_y)$. The normalized time average over an interval $\tau$ is denoted by:

$$\overline{Z(t)}^\tau = \frac{1}{\tau\langle Z(t)\rangle}\int_0^\tau Z(t+\tau')d\tau'. \tag{23}$$

When the interval $\tau$ is lager than the characteristic time of the system, the FT holds when:

$$S_{\overline{Z}^\tau}(z) = \sigma\tau z, \tag{24}$$

where the variable $\sigma$ is called the contraction rate (see Ciliberto et al. (2004)), it depends on the problem considered.

The power the atmosphere looses at the interface $P_{a\to i;\omega}$ and the power the ocean receives from the atmosphere at the interface $P_{i\to o;\omega}$ along a trajectory $\omega\in\Omega$ is investigated. Both differ by the work dissipated at the interface (see section 5). The ensemble averages of all these quantities are positive, but negative fluxes exist, even when temporal averages over time intervals of length $\tau$ are considered. The FT states that the probability of finding a positive flux of magnitude $z$ divided by the probability of a negative flux with the same magnitude increases exponentially with the value $z$ and the averaging period $\tau$.

In the problem considered here the variable $\overline{Z(t)}^\tau$ is either the time-averaged energy the atmosphere does on the ocean $\overline{P_{a\to i;\omega}}^\tau$, divided by its ensemble average, or the time-averaged work the ocean receives from the atmosphere $\overline{P_{i\to o;\omega}}^\tau$, divided by its ensemble average. More precisely, the random variables:

$$\overline{Z_{a;\omega}(t)}^\tau = \frac{\overline{P_{a\to i;\omega}(t)}^\tau}{\langle P_{a\to i;\omega}(t)\rangle_\Omega} \qquad \text{and} \qquad \overline{Z_{o;\omega}(t)}^\tau = \frac{\overline{P_{i\to o;\omega}(t)}^\tau}{\langle P_{i\to o;\omega}(t)\rangle_\Omega}, \tag{25}$$

for all the models are considered. Ensemble averages of the fluxes can be obtained analytically for the linear models, but I do not know their pdfs. These investigations are therefore numerical even for the linear models considered here.

First, the L3 model is discussed. It is important to note that although $\langle P_{a\to i;\omega}(t)\rangle_\Omega$ and $\langle P_{i\to o;\omega}(t)\rangle_\Omega$ are constant in time (after an initial spin-up of $O((SM)^{-1})$, see Wirth (2018)) the pdfs are not (see Fig 2). Which means that the energy transfers are

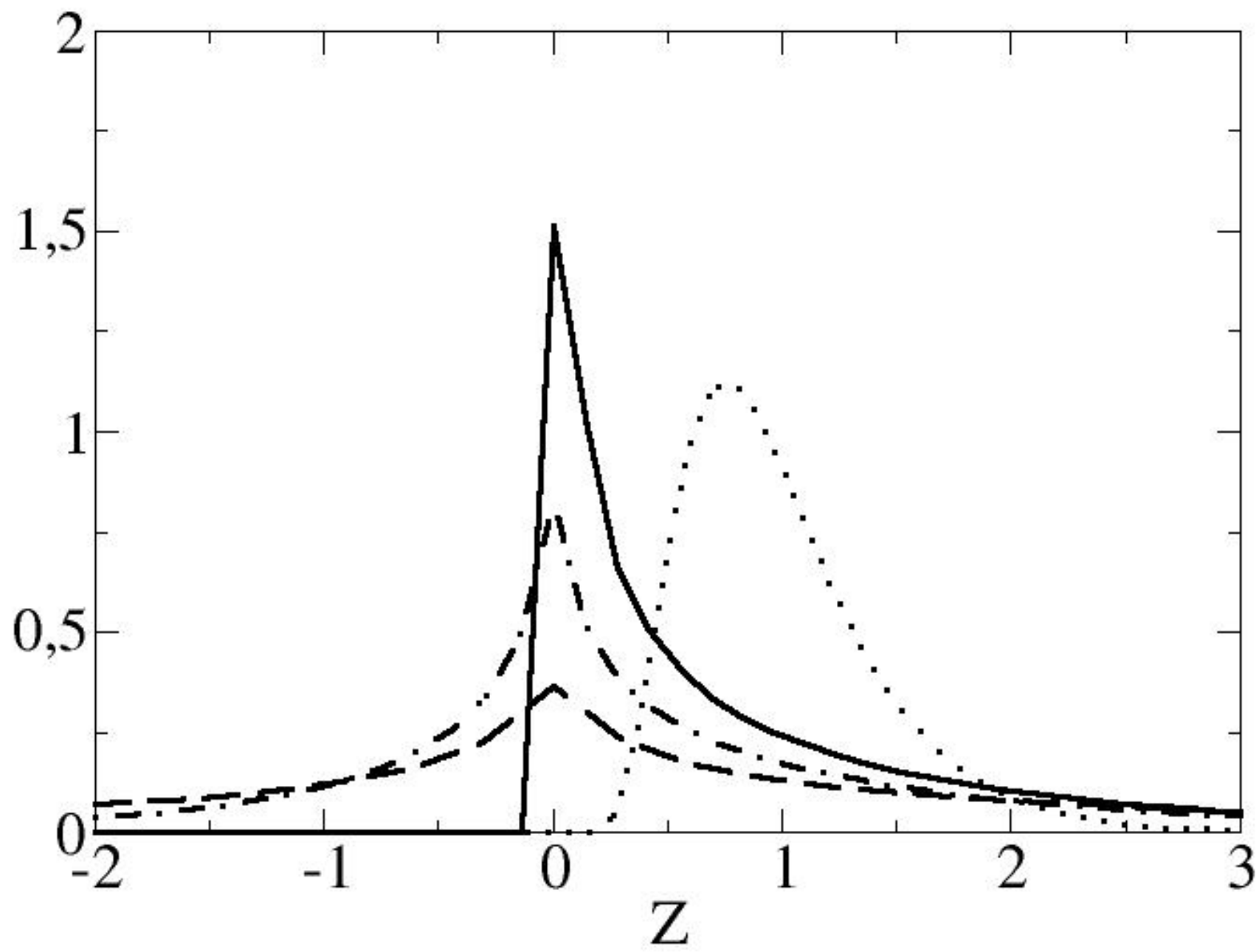

**Figure 2.** Probability density function of $\overline{Z_{a/o;\omega}(t)}^{\tau}$ at $t = 300$ and $\tau = 0$, atmosphere full-line, ocean dashed-line and for $\tau = 100$, atmosphere dotted-line, ocean dashed-dotted-line. All are clearly non-Gaussian

non-stationary processes with a constant-in-time average value. More precisely: the pdfs of the variables $\overline{Z_{a;\omega}}^{\tau}$ and $\overline{Z_{o;\omega}(t)}^{\tau}$ depend on $t$ and $\tau$, but $\langle\overline{Z_{a;\omega}(t)}^{\tau}\rangle_\Omega$ and $\langle\overline{Z_{o;\omega}(t)}^{\tau}\rangle_\Omega$ are independent of the time $t$ and the averaging period $\tau$.

The parameters used in the numerical calculations are $S = 10^{-3}$ and $m = 100$. Details on solving numerically the stochastic differential equations are given in Wirth (2018). The numerical results presented in fig. 2 show that the pdfs of $\overline{Z_{a;\omega}(t)}^{\tau}$ and $\overline{Z_{o;\omega}(t)}^{\tau}$ for $t = 300$, and $\tau = 0, 100$ are non Gaussian. The exponential scaling of the symmetry function for the ocean $S_{\overline{Z_o}^{\tau}}(z)$ for $t = 300$, and $\tau = 0, 100, 200$ is clearly present in fig. 3. This also means that zero is a special value, which is already conspicuous in fig 2. The scaling exponents for $t = 10, 20, 30$ and 50 as a function of $\tau$ are given in fig. 4, it can be verified that eq. (24) holds, when the absolute and the averaging time exceeds the characteristic time $t, \tau > (SM)^{-1}$ meaning that the FT applies asymptotically, as in Ciliberto et al. (2004). The change of slope for the different values of $t$ is well fitted by a $\sigma \propto t^{-1}$ law.

For the atmosphere the probability of having a negative flux $P_{a\to i}(t)$, that is the atmosphere receives energy at the interface due to the ocean dynamics is small even in instantaneous pdfs. Negative events in an ensemble sizes of $3 \cdot 10^7$ were so few that the symmetry function could not be obtained with a sufficient accuracy.

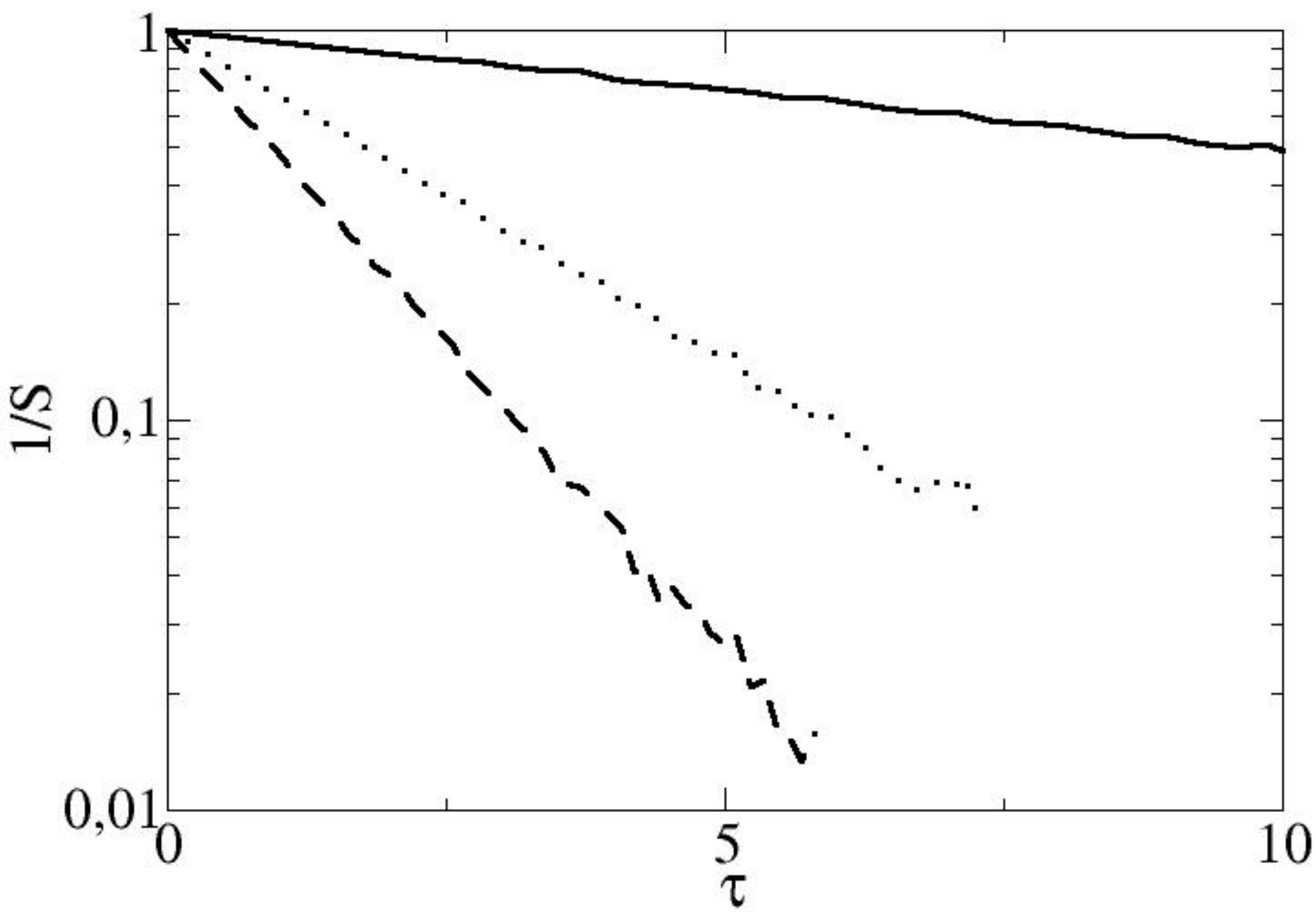

**Figure 3.** Lin-log plot of $(S_{\overline{Z_o}^\tau})^{-1}$ for $t = 300$, $\tau = 0$ full-line, 10 dotted-line, 20 dashed-line

For calculations with the colored noise model, the same parameters as in the white-noise calculations are used and $\mu = 10^{-2}$. In this case the forcing time-scale $\mu^{-1}$ is actually slower than the atmospheric friction time-scale $(Sm)^{-1}$ but faster than the oceanic friction time-scale $S^{-1}$. This mimics the fact, that the fast motion in the atmospheric boundary-layer is forced by the slower synoptic dynamics above. The forcing time of the oceanic mixed layer is then determined by the mass ratio $m$ between the oceanic and the atmospheric layers, it is the slowest time-scale. Results (not shown) from the models with colored noise agree qualitatively with the white-noise forcing, that is, they indicate that the energy flux to the ocean obeys a FT.

Numerical integration of the L1W, L2W, L1C and L2C models show that $P_{i\to o;\omega}$ obeys the FT as in the L3 model. The atmospheric flux $P_{a\to i}(t)$ is always positive in the L1 and L2 models so the FT can not be considered.

## 8 Discussion

When ocean velocities are not considered in models of air sea-interaction the atmosphere loses, on average, more energy and the ocean gains more energy, as when the ocean velocities are taken into account.

Previous publications on the comparison of different models of air-sea interaction focus on quantitative differences. This is justified when the short-term dynamics is considered, as shown above. At longer time-scales the differences are not only

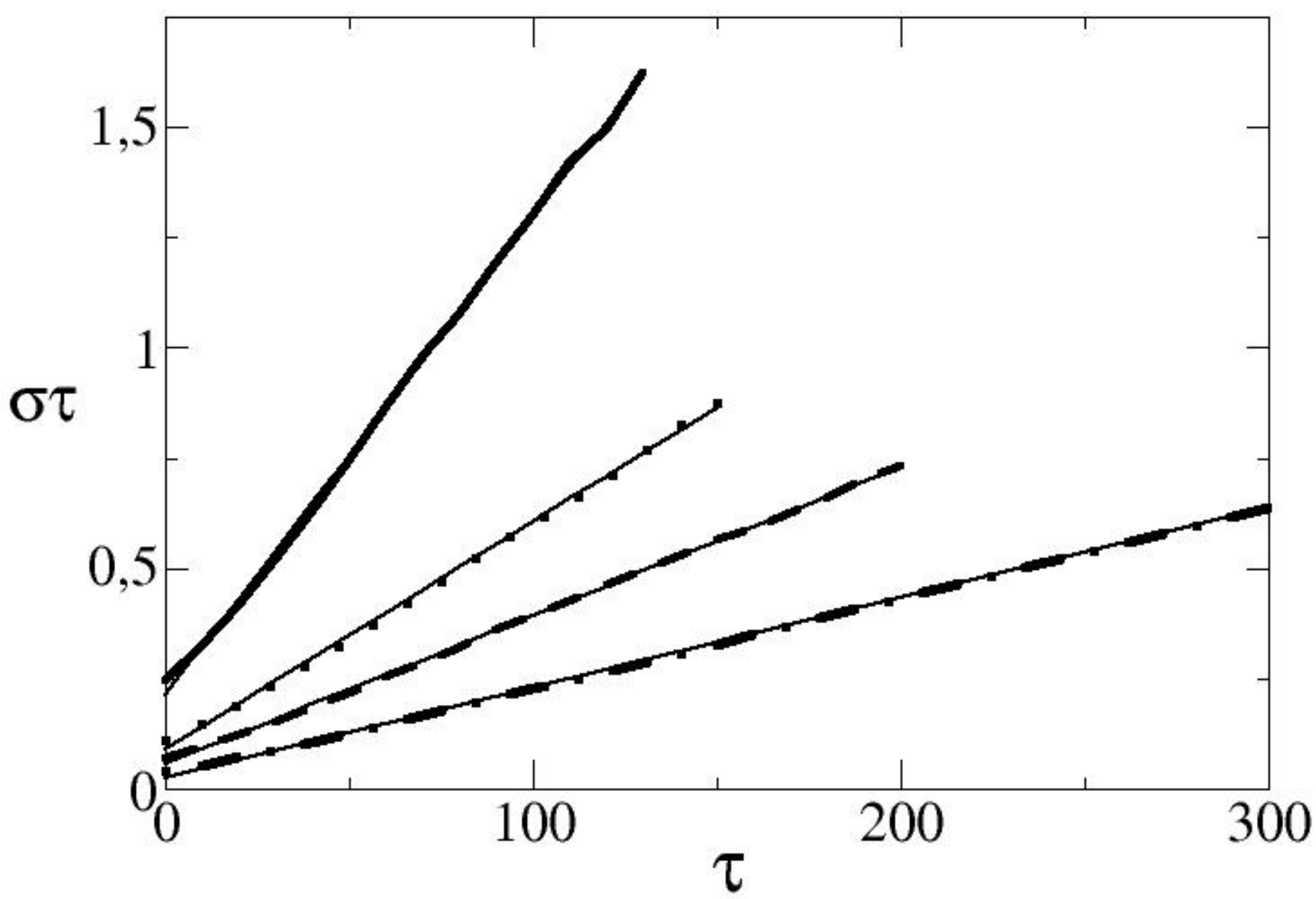

**Figure 4.** Scaling exponents for $t = 10$ full-line, 20 dotted-line, 30 dashed-line and 50 dashed-dotted-line, as a function of $\tau$ and their linear fit (thin full lines), the slope is $\sigma$.

quantitative but qualitative, as for some models stationary states in the ocean the atmosphere or in both are reached while in others this is not the case. An example is the "eddy-killing" term (see Renault et al. (2017)), that includes the ocean velocity in the shear calculation. In the short term its impact is small and can be parameterized by changing the friction coefficient. In the long-term it imposes a convergence to a stationary state, when only implemented in the ocean, when implemented in the atmosphere and the ocean it leads to a divergence in both layers, whereas neglecting it totally leads to a divergence in the ocean only. In more involved models, divergence is avoided by other processes as non-linear interactions or data assimilation, who assure it differently. When these processes supplante an incomplete representation of "eddy killing" the converged state will differ and so will the energy balance. The small differences in the short term behavior and the qualitative differences in the long-term behavior between the models, indicate that the choice of the model does not matter when weather or ocean forecasts are performed, but it might be crucial in climate simulations.

The discussion of the FDT establishes when the response to an external perturbation can be obtained from internal fluctuations of the system. In the simple system discussed here we can see analytically when it is verified and fails and how the failure can be removed by extending the phase space. To determine the response to a sudden change in the external forcing is key in many applications, as for example the response of the atmospheric and oceanic planetary-boundary-layer dynamics to

a change in the synoptic weather condition. The presented calculations can also be used to guide applications of the FDT to systems with large, but not infinite, time separation.

The FT concerns the transfer of energy between the atmosphere and the interface and the interface and the ocean on different time scales. The temporal down-scaling is solved when we can obtain the pdf of short term averages from the pdf of longer term averages. The temporal up-scaling is solved when we can obtain the pdf of long term averages from the pdf of short term averages. The FT relates temporal averages over different time scales and puts a large constraint on the probability density functions (pdfs) of the averaged energy transfers over different time scales. The FT is key to understanding and modeling the climate dynamics as in all observations and models some time-and-space averaging is present. It is not even clear what is the averaging period associated to a variable in a model. The FT gives us a hint of what to expect when passing, for example, from monthly-averaged interaction / forcing to daily or hourly averages.

In Renault et al. (2017) it was suggested, using satellite observations, that to leading order, the effect of mesoscale ocean currents on the surface stress (the "eddy killing") can be parameterized as a linear function of the wind. Throughout the above presented results we see that differences between the three models considered, are not only quantitative but are qualitative, when long-term behavior is considered. This shows that we can not improve the L1 or L2 model by adjusting the friction parameter to obtain the behavior of the consistent L3 model.

The major difference between a two-dimensional and a local model is that the former contains horizontal advection of momentum while the latter does not. It is thus not clear which variable of the two-dimensional model has to be considered using the insight from the local models; is it the local velocity, the velocity advected by the total inertia mode or by the ocean dynamics. Or do we have to consider coarse grained variables for which the importance of horizontal advection is reduced? If this is the case we have to define a coarse graining scale that is sufficient or optimal in some sense.

## 9 Conclusions

In the present work deterministic and stochastic calculus to established models of momentum transfer at the air-sea interface are considered. The results of these idealized models have implications for present day simulations of climate dynamics and are important to understand the mechanical energy transfer between the ocean and the atmosphere.

It is furthermore interesting to see the concepts of non-equilibrium statistical dynamics applied to a field of climate science which is sufficiently simple so that most results can be obtained analytically. This, together with the detailed calculations given in the appendixes, will further the advancement of this concepts in the field.

After having here considered linear 0D models, which allow for analytic calculations to a large extent, future work will be dedicated to extend these work to more involved, non-linear, 1D, 2D and 3D numerical models and to consider the FDR, FDT and FT in observations.

The here presented theory is not restricted to momentum transfer, but can also be employed to study heat exchange between the atmosphere and the ocean, or to other processes in the climate system with diverse characteristic time scales.

In this work basic models of air-sea interaction are investigated using modern concepts of non-equilibrium statistical dynamics. In the sense of Kuhn (1963) this work is “normal science”, puzzle solving within an established framework. It is important to be familiar with these calculations and and their results because they demonstrate how far analytic or semi-analytic calculus brings us and they also help us to understand observations. If discrepancy between the former and the latter arises, which can not be solved, they possibly lead to a paradigm shift. But for this to happen the science within a certain paradigm has to be scrutinized (see again Kuhn (1963)). Only after this is accomplished deviations from the theory, as a glass transition in the air-sea interaction framework discovered in Moulin and Wirth (2016) can be studied.

*Code availability.* The data used in section 7 was produced by the FORTRAN code available under open acces under: https://zenodo.org/record/2530007

## Appendix A: The Models

In this section the solution of the linear models L1, L2 and L3 are solved using linear algebra. The linear differential equation

$$\partial_t u \quad = \quad -\alpha u + F(t) \tag{A1}$$

with initial condition $u(t_0)$, has solutions:

$$u(t) = \mathcal{I}_{t_0}^{t}(\alpha) + u(t_0)e^{\alpha(t_0-t)}, \tag{A2}$$

with the symbol:

$$\mathcal{I}_{t_0}^{t}(\alpha) = \int_{t_0}^{t} e^{\alpha(t'-t)} F(t')dt'. \tag{A3}$$

In the multidimensional case we have:

$$\partial_t \boldsymbol{u} \quad = \quad -P\boldsymbol{u} + \boldsymbol{F}F(t) = -ADA^{-1} + \boldsymbol{F}F(t) \tag{A4}$$

where $D$ is a diagonal (or Jacobi normal) matrix, $\boldsymbol{F}$ a constant coefficient vector and $F(t)$ a time dependent scalar forcing. the solution with initial condition $\boldsymbol{u}(t_0)$ is:

$$\boldsymbol{u}(t) \quad = \quad A\mathcal{I}_{t_0}^{t}(D)A^{-1}\boldsymbol{F} + Ae^{D(t_0-t)}A^{-1}\boldsymbol{u}(t_0). \tag{A5}$$

Note that in all our applications $\alpha$ and the eigenvalues of $D$ are positive or zero.

### A1 Model L1

The system is forced and damped and the atmospheric dynamics acts on the ocean without considering the ocean velocity. A coupling which is still used in some climate models.

$$\partial_t u_a \quad = \quad -Smu_a + F_a \tag{A6}$$

$$\partial_t u_o \quad = \quad Su_a + F_o \tag{A7}$$

$$\partial_t \begin{pmatrix} u_a \\ u_o \end{pmatrix} = \begin{pmatrix} -Sm & 0 \\ S & 0 \end{pmatrix} \begin{pmatrix} u_a \\ u_o \end{pmatrix} + \begin{pmatrix} F_a \\ F_o \end{pmatrix} \tag{A8}$$

$$\lambda_1 = -Sm,\ e_1 = \begin{pmatrix} m \\ -1 \end{pmatrix};\ \lambda_2 = 0,\ e_2 = \begin{pmatrix} 0 \\ 1 \end{pmatrix} \tag{A9}$$

$$P = ADA^{-1} = \begin{pmatrix} -Sm & 0 \\ S & 0 \end{pmatrix} = \begin{pmatrix} m & 0 \\ -1 & 1 \end{pmatrix} \begin{pmatrix} -Sm & 0 \\ 0 & 0 \end{pmatrix} \begin{pmatrix} m^{-1} & 0 \\ m^{-1} & 1 \end{pmatrix}. \tag{A10}$$

The solution is:

$$\begin{pmatrix} u_a(t) \\ u_o(t) \end{pmatrix} = A\mathcal{I}_{t_0}^t(D)A^{-1} \begin{pmatrix} F_a \\ F_o \end{pmatrix} = A \begin{pmatrix} \mathcal{I}_{t_0}^t(Sm) & 0 \\ 0 & \mathcal{I}_{t_0}^t(0) \end{pmatrix} A^{-1} \begin{pmatrix} F_a \\ F_o \end{pmatrix}$$
$$= \begin{pmatrix} \mathcal{I}_{t_0}^t(Sm) & 0 \\ \left[-\mathcal{I}_{t_0}^t(Sm) + \mathcal{I}_{t_0}^t(0)\right]/m & \mathcal{I}_{t_0}^t(0) \end{pmatrix} \begin{pmatrix} F_a \\ F_o \end{pmatrix}. \tag{A11}$$

In the absence of forcing an initial perturbation at $\Delta t = 0$, $(\bar{u}_a, \bar{u}_o)$ evolves as:

$$\begin{pmatrix} u_a^P(\Delta t) \\ u_o^P(\Delta t) \end{pmatrix} = \begin{pmatrix} e^{-Sm\Delta t} & 0 \\ \frac{1-e^{-Sm\Delta t}}{m} & 1 \end{pmatrix} \begin{pmatrix} \bar{u}_a \\ \bar{u}_o \end{pmatrix}. \tag{A12}$$

Note that the perturbation matrix above: $\chi(\Delta t) = A\exp(Dt)A^{-1}$, is independent of the time $t$.

## A2 Model L2

The system is forced with damping and the atmospheric dynamics forces the ocean, ocean velocity is taken into account for the ocean dynamics but not in the atmospheric dynamics (Newton's third law is not respected).

$$\partial_t u_a = -Smu_a + F_a \tag{A13}$$

$$\partial_t u_o = S(u_a - u_o) + F_o \tag{A14}$$

$$\partial_t \begin{pmatrix} u_a \\ u_o \end{pmatrix} = \begin{pmatrix} -Sm & 0 \\ S & -S \end{pmatrix} \begin{pmatrix} u_a \\ u_o \end{pmatrix} + \begin{pmatrix} F_a \\ F_o \end{pmatrix} \tag{A15}$$

$$\lambda_1 = -Sm,\ e_1 = \begin{pmatrix} m-1 \\ -1 \end{pmatrix};\ \lambda_2 = -S,\ e_2 = \begin{pmatrix} 0 \\ 1 \end{pmatrix} \tag{A16}$$

$$P = ADA^{-1} = \begin{pmatrix} -Sm & 0 \\ S & -S \end{pmatrix} = \begin{pmatrix} m-1 & 0 \\ -1 & 1 \end{pmatrix} \begin{pmatrix} -Sm & 0 \\ 0 & -S \end{pmatrix} \begin{pmatrix} (m-1)^{-1} & 0 \\ (m-1)^{-1} & 1 \end{pmatrix}. \tag{A17}$$

The solution is:

$$\begin{pmatrix} u_a(t) \\ u_o(t) \end{pmatrix} = A\mathcal{I}_{t_0}^t(D)A^{-1}\begin{pmatrix} F_a \\ F_o \end{pmatrix}$$

$$\begin{pmatrix} u_a(t) \\ u_o(t) \end{pmatrix} = A\begin{pmatrix} \mathcal{I}_{t_0}^t(Sm) & 0 \\ 0 & \mathcal{I}_{t_0}^t(S) \end{pmatrix} A^{-1}\begin{pmatrix} F_a \\ F_o \end{pmatrix}$$

$$= \begin{pmatrix} \mathcal{I}_{t_0}^t(Sm) & 0 \\ \left[-\mathcal{I}_{t_0}^t(Sm) + \mathcal{I}_{t_0}^t(S)\right]/(m-1) & \mathcal{I}_{t_0}^t(S) \end{pmatrix} \begin{pmatrix} F_a \\ F_o \end{pmatrix}. \tag{A18}$$

In the absence of forcing an initial perturbation at $\Delta t = 0$, $(\bar{u}_a, \bar{u}_o)$ evolves as:

$$\begin{pmatrix} u_a^P(\Delta t) \\ u_o^P(\Delta t) \end{pmatrix} = \begin{pmatrix} e^{-Sm\Delta t} & 0 \\ \frac{e^{-S\Delta t} - e^{-Sm\Delta t}}{m-1} & e^{-S\Delta t} \end{pmatrix} \begin{pmatrix} \bar{u}_a \\ \bar{u}_o \end{pmatrix}. \tag{A19}$$

Note that the perturbation matrix above: $\chi(\Delta t) = A\exp(Dt)A^{-1}$, is independent of the time $t$.

### A3 Model L3

The atmosphere is forced with damping and the atmospheric dynamics forces the ocean, the ocean velocity is taken into account.

$$\partial_t u_a = Sm(u_o - u_a) + F_a \tag{A20}$$

$$\partial_t u_o = S(u_a - u_o) + F_o \tag{A21}$$

$$\partial_t \begin{pmatrix} u_a \\ u_o \end{pmatrix} = \begin{pmatrix} -Sm & Sm \\ S & -S \end{pmatrix} \begin{pmatrix} u_a \\ u_o \end{pmatrix} + \begin{pmatrix} F_a \\ F_o \end{pmatrix} \tag{A22}$$

$$\lambda_1 = -SM,\ e_1 = \begin{pmatrix} m \\ -1 \end{pmatrix};\ \lambda_2 = 0,\ e_2 = \begin{pmatrix} 1 \\ 1 \end{pmatrix}. \tag{A23}$$

The first eigenvector corresponds to the shear mode and the second to the total-inertia mode.

$$P = ADA^{-1} = \begin{pmatrix} -Sm & Sm \\ S & -S \end{pmatrix} = \begin{pmatrix} m & 1 \\ -1 & 1 \end{pmatrix} \begin{pmatrix} -SM & 0 \\ 0 & 0 \end{pmatrix} \frac{1}{M} \begin{pmatrix} 1 & -1 \\ 1 & m \end{pmatrix}. \tag{A24}$$

The solution is:

$$\begin{pmatrix} u_a(t) \\ u_o(t) \end{pmatrix} = A\mathcal{I}_{t_0}^t(D)A^{-1}\begin{pmatrix} F_a \\ F_o \end{pmatrix}$$

$$\begin{pmatrix} u_a(t) \\ u_o(t) \end{pmatrix} = A\begin{pmatrix} \mathcal{I}_{t_0}^t(SM) & 0 \\ 0 & \mathcal{I}_{t_0}^t(0) \end{pmatrix}A^{-1}\begin{pmatrix} F_a \\ F_o \end{pmatrix}$$

$$= \frac{1}{M}\begin{pmatrix} m\mathcal{I}_{t_0}^t(SM)+\mathcal{I}_{t_0}^t(0) & -m\mathcal{I}_{t_0}^t(SM)+m\mathcal{I}_{t_0}^t(0) \\ -\mathcal{I}_{t_0}^t(SM)+\mathcal{I}_{t_0}^t(0) & \mathcal{I}_{t_0}^t(SM)+m\mathcal{I}_{t_0}^t(0) \end{pmatrix}\begin{pmatrix} F_a \\ F_o \end{pmatrix}. \tag{A25}$$

In the absence of forcing an initial perturbation at $\Delta t = 0$, $(\bar{u}_a, \bar{u}_o)$ evolves as:

$$\begin{pmatrix} u_a^P(\Delta t) \\ u_o^P(\Delta t) \end{pmatrix} = \frac{1}{M}\begin{pmatrix} 1+me^{-SM\Delta t} & m(1-e^{-SM\Delta t}) \\ 1-e^{-SM\Delta t} & m+e^{-SM\Delta t} \end{pmatrix}\begin{pmatrix} \bar{u}_a \\ \bar{u}_o \end{pmatrix}. \tag{A26}$$

Note that the perturbation matrix above: $\chi(\Delta t) = A\exp(Dt)A^{-1}$, is independent of the time $t$.

## Appendix B: Experiments

In all experiments only the atmosphere is forced, $F_o = 0$. Note that in the L3 model the atmosphere and the ocean are treated similarly and they only differ by the mass ratio $m$. The dynamics due to a forcing of the ocean can be represented by choosing $m < 1$ and interchanging the subscripts. The dynamics of a forcing of the ocean and atmosphere can be obtained by adding a model forced by the ocean and the same model forced by the atmosphere due to the linearity of the model.

### B1 Constant Forcing

In this appendix $F_a = 1$ and $F_o = 0$. Put into eq. (A3) this leads to:

$$\mathcal{I}_{t_0}^t(0) = t - t_0 \tag{B1}$$

$$\mathcal{I}_{t_0}^t(\alpha) = \frac{1}{\alpha}(1-\exp(-\alpha(t-t_0))), \text{ if } \alpha \neq 0. \tag{B2}$$

The solutions for the different models are:

### B2 Experiment L1K

$$\begin{pmatrix} u_a(t) \\ u_o(t) \end{pmatrix} = \frac{1}{Sm}\begin{pmatrix} 1-\exp(-Sm(t-t_0)) \\ [Sm(t-t_0)-1+\exp(-Sm(t-t_0))]/m \end{pmatrix}. \tag{B3}$$

The Taylor series expansion for small times $(t-t_0) \ll (Sm)^{-1}$ is:

$$\begin{pmatrix} u_a(t) \\ u_o(t) \end{pmatrix} = \begin{pmatrix} t-t_0-\frac{Sm}{2}(t-t_0)^2+\frac{(Sm)^2}{6}(t-t_0)^3-... \\ \frac{S}{2}(t-t_0)^2-\frac{S^2m}{6}(t-t_0)^3+\frac{S^3m^2}{24}(t-t_0)^4-... \end{pmatrix}. \tag{B4}$$

Open Access EGU

Asymptotics for large times $(t-t_0) \gg (Sm)^{-1}$:

$$\begin{pmatrix} u_a(t) \\ u_o(t) \end{pmatrix} = \begin{pmatrix} \frac{1}{Sm} \\ \frac{t-t_0}{m} - \frac{1}{Sm^2} \end{pmatrix} \tag{B5}$$

### B3 Experiment L2K

$$\begin{pmatrix} u_a(t) \\ u_o(t) \end{pmatrix} = \begin{pmatrix} [1-\exp(-Sm(t-t_0))]/(Sm) \\ [1-\exp(-S(t-t_0)) - (1-\exp(-Sm(t-t_0))/m]/(S(m-1)) \end{pmatrix}. \tag{B6}$$

The Taylor series expansion for small times $(t-t_0) \ll (Sm)^{-1}$ is:

$$\begin{pmatrix} u_a(t) \\ u_o(t) \end{pmatrix} = \begin{pmatrix} t-t_0 - \frac{Sm}{2}(t-t_0)^2 + \frac{(Sm)^2}{6}(t-t_0)^3 - \dots \\ \frac{S}{2}(t-t_0)^2 - \frac{S^2M}{6}(t-t_0)^3 + \frac{S^3(m^2+m+1)}{24}(t-t_0)^4 - \dots \end{pmatrix}. \tag{B7}$$

Asymptotics for large times $(t-t_0) \gg S^{-1}$:

$$\begin{pmatrix} u_a(t) \\ u_o(t) \end{pmatrix} = \begin{pmatrix} \frac{1}{Sm} \\ \frac{1}{Sm} \end{pmatrix}. \tag{B8}$$

### B4 Experiment L3K

$$\begin{pmatrix} u_a(t) \\ u_o(t) \end{pmatrix} = \frac{1}{M} \begin{pmatrix} t-t_0 + (1-\exp(-SM(t-t_0)))m/(SM) \\ t-t_0 - (1-\exp(-SM(t-t_0)))/(SM) \end{pmatrix} \tag{B9}$$

The Taylor series expansion for small times $(t-t_0) \ll (Sm)^{-1}$ is:

$$\begin{pmatrix} u_a(t) \\ u_o(t) \end{pmatrix} = \begin{pmatrix} t-t_0 - \frac{Sm}{2}(t-t_0)^2 + \frac{S^2mM}{6}(t-t_0)^3 - \dots \\ \frac{S}{2}(t-t_0)^2 - \frac{S^2M}{6}(t-t_0)^3 + \frac{S^3M^2}{24}(t-t_0)^4 - \dots \end{pmatrix} \tag{B10}$$

Asymptotics for large times $(t-t_0) \gg (SM)^{-1}$:

$$\begin{pmatrix} u_a(t) \\ u_o(t) \end{pmatrix} = \begin{pmatrix} \frac{t-t_0}{M} + \frac{m}{SM^2} \\ \frac{t-t_0}{M} - \frac{1}{SM^2} \end{pmatrix} \tag{B11}$$

### B5 Periodic Forcing

In this appendix $F_a = \cos(\kappa t)$. Put into eq. (A3) and starting the integration at $t_0 = -\infty$ (ignoring transients), leads to:

$$\mathcal{I}^t_{-\infty}(\alpha) = \frac{1}{\alpha^2+\kappa^2}(\kappa\sin(\kappa t) + \alpha\cos(\kappa t)). \tag{B12}$$

## B6 Experiment L1P

Solution

$$\begin{pmatrix} u_a(t) \\ u_o(t) \end{pmatrix} = \frac{1}{(Sm)^2+\kappa^2} \begin{pmatrix} (\kappa\sin(\kappa t) + Sm\cos(\kappa t)) \\ S/\kappa(Sm\sin(\kappa t) - \kappa\cos(\kappa t)) \end{pmatrix}. \tag{B13}$$

The averages over one period $\tau = 2\pi/\kappa$ are denoted by $\langle . \rangle_\tau$. First order moments all vanish, for the second order moments we get:

$$\begin{aligned} 2\langle u_a^2 \rangle_\tau &= \frac{1}{(Sm)^2+\kappa^2} \\ 2\langle u_o^2 \rangle_\tau &= \frac{S^2}{\kappa^2((Sm)^2+\kappa^2)} \\ \frac{\langle u_o^2 \rangle_\tau}{\langle u_a^2 \rangle_\tau} &= \frac{S^2}{\kappa^2} \\ \langle u_a u_o \rangle_\tau &= 0. \end{aligned} \tag{B14}$$

## B7 Experiment L2P

$$u_a = \frac{1}{(Sm)^2+\kappa^2}[\kappa\sin(\kappa t) + Sm\cos(\kappa t)] \tag{B15}$$

$$u_o = \frac{S}{((Sm)^2+\kappa^2)(S^2+\kappa^2)}[SM\kappa\sin(\kappa t) + (S^2m - \kappa^2)\cos(\kappa t)] \tag{B16}$$

$$\begin{aligned} 2\langle u_a^2 \rangle_\tau &= \frac{1}{(Sm)^2+\kappa^2} \\ 2\langle u_o^2 \rangle_\tau &= 2\langle u_a u_o \rangle_\tau = \frac{S^2}{((Sm)^2+\kappa^2)(S^2+\kappa^2)} \\ \frac{\langle u_o^2 \rangle_\tau}{\langle u_a^2 \rangle_\tau} &= \frac{S^2}{S^2+\kappa^2} \end{aligned}$$

## B8 Experiment L3P

$$\begin{aligned} u_t(t) &= \frac{\sin(\kappa t)}{\kappa} \\ u_s(t) &= \frac{\kappa\sin(\kappa t) + SM\cos(\kappa t)}{\kappa^2 + (SM)^2} \end{aligned} \tag{B17}$$

$$\begin{aligned} u_a(t) &= \frac{1}{M}(u_t + mu_s) = \frac{1}{\kappa^2+(SM)^2}\left[\frac{\kappa^2+S^2M}{\kappa}\sin(\kappa t) + Sm\cos(\kappa t)\right] \\ u_o(t) &= \frac{1}{M}(u_t - u_s) = \frac{S}{\kappa(\kappa^2+(SM)^2)}[SM\sin(\kappa t) - \kappa\cos(\kappa t)] \end{aligned} \tag{B18}$$

which leads to:

$$2\langle u_a^2\rangle_\tau = \frac{1}{\kappa^2+(SM)^2}\frac{\kappa^2+S^2}{\kappa^2} \tag{B19}$$

$$2\langle u_o^2\rangle_\tau = 2\langle u_a u_o\rangle_t = \frac{1}{\kappa^2+(SM)^2}\frac{S^2}{\kappa^2}. \tag{B20}$$

$$\frac{\langle u_o^2\rangle_\tau}{\langle u_a^2\rangle_\tau} = \frac{S^2}{\kappa^2+S^2}. \tag{B21}$$

### B9 Stochastic calculus

### B10 Random walk and Ornstein-Uhlenbeck process

The following identities are used:

$$\int_{-\infty}^{t} \delta(t-t')dt' = 1/2 \tag{B22}$$

$$\int_{t_0}^{t}\int_{t_0}^{t} \delta(t''-t')dt''dt' = t-t_0 \tag{B23}$$

$$\langle F_\omega(t)\rangle_\Omega = 0 \tag{B24}$$

$$\langle F_\omega(t')F_\omega(t'')\rangle_\Omega = 2R\delta(t''-t'). \tag{B25}$$

In the sequel the subscript $\omega$ is omitted. Below are the equations for a random-walk $u_R$ and a Ornstein-Uhlenbeck process $u_O$, the solution of a Langevin equation.

$$\partial_t u_R = F \tag{B26}$$

$$\partial_t u_O = -Su_O + F. \tag{B27}$$

Solutions starting from rest at $t_0$, $u_R(t_0) = u_O(t_0) = 0$, are:

$$u_R(t) = \int_{t_0}^{t} F(t')dt' \tag{B28}$$

$$u_O(t) = \int_{t_0}^{t} e^{S(t'-t)}F(t')dt'. \tag{B29}$$

It follows that: $\langle u_R\rangle_\Omega = \langle u_O\rangle_\Omega = 0$. Second order moments are (note that as processes are Gaussian first and second order moments completely determine the stochastic processes):

$$\begin{aligned}
\langle u_R^2(t)\rangle_\Omega &= \int_{t_0}^{t}\int_{t_0}^{t} \langle F(t')F(t'')\rangle_\Omega dt''dt' \\
&= 2R(t-t_0) \qquad \text{(B30)}\\
\langle u_R(t)u_O(t)\rangle_\Omega &= \int_{t_0}^{t}\int_{t_0}^{t} e^{S(t'-t)}\langle F(t')F(t'')\rangle_\Omega dt''dt' \\
&= \frac{2R}{S}(1-e^{S(t_0-t)}) \qquad \text{(B31)}\\
\langle u_O^2(t)\rangle_\Omega &= \int_{t_0}^{t}\int_{t_0}^{t} e^{S(t'+t''-2t)}\langle F(t')F(t'')\rangle_\Omega dt''dt' \\
&= \frac{R}{S}(1-e^{2S(t_0-t)}). \qquad \text{(B32)}
\end{aligned}$$

$$\mathcal{I}_{t_0}^{t}(\alpha) = \int_{t_0}^{t} e^{\alpha(t'-t)}F(t')dt'. \qquad \text{(B33)}$$

$$\begin{aligned}
\langle \mathcal{I}_{t_0}^{t}(\alpha)\mathcal{I}_{t_0}^{t+\Delta t}(\beta)\rangle_\Omega &= 2R(t-t_0)e^{-\beta\Delta t} \text{ if } \alpha+\beta=0 \qquad \text{(B34)}\\
\langle \mathcal{I}_{t_0}^{t}(\alpha)\mathcal{I}_{t_0}^{t+\Delta t}(\beta)\rangle_\Omega &= \frac{2R}{\alpha+\beta}(1-\exp(-(\alpha+\beta)(t-t_0))e^{-\beta\Delta t} \text{ if } \alpha+\beta\neq 0. \qquad \text{(B35)}
\end{aligned}$$

Also note that in all cases:

$$\langle \mathcal{I}_{t_0}^{t}(\alpha)\mathcal{I}_{t_0}^{t+\Delta t}(\beta)\rangle_\Omega = \langle \mathcal{I}_{t_0}^{t}(\alpha)\mathcal{I}_{t_0}^{t}(\beta)\rangle_\Omega e^{-\beta\Delta t}. \qquad \text{(B36)}$$

The correlation matrix ($t$-dependence is kept to deal with non-stationary processes) is

$$\begin{aligned}
C(t,\Delta t) &= \langle \boldsymbol{u}(t+\Delta t)\cdot \boldsymbol{u}(t)^T\rangle_\Omega \\
&= A\langle \mathcal{I}_{t_0}^{t+\Delta t}(D)A^{-1}\boldsymbol{F}\cdot\boldsymbol{F}^T(A^{-1})^T\mathcal{I}_{t_0}^{t}(D)\rangle_\Omega A^T \\
&= Ae^{-D\Delta t}\langle \mathcal{I}_{t_0}^{t}(D)A^{-1}\boldsymbol{F}\cdot\boldsymbol{F}^T(A^{-1})^T\mathcal{I}_{t_0}^{t}(D)\rangle_\Omega A^T, \qquad \text{(B37)}
\end{aligned}$$

where we used eqs. (A5) and (B36). Even so the matrix $\boldsymbol{F}\cdot\boldsymbol{F}^T$ is singular, this does not necessarily lead to a singular correlation matrix, as $\langle a\rangle\langle b\rangle \neq \langle ab\rangle$. Calculations show that the normalized correlation matrix $C(t,\Delta t)C(t,0)^{-1}$ is independent of $\boldsymbol{F}$, if none of the eigenvectors is orthogonal to $\boldsymbol{F}$, that is all eigenvectors are subject to the forcing. To see this we transform to a

coordinate system spanned by the eigenvectors, in this case $P = D$ is diagonal and $A$ is the identity matrix, then:

$$\begin{aligned} C(t,\Delta t)C(t,0)^{-1} &= A\langle \mathcal{I}_{t_0}^{t+\Delta t}(D)A^{-1}\boldsymbol{F}\cdot\boldsymbol{F}^T(A^{-1})^T\mathcal{I}_{t_0}^{t}(D)\rangle_\Omega \\ &\quad (\langle \mathcal{I}_{t_0}^{t}(D)A^{-1}\boldsymbol{F}\cdot\boldsymbol{F}^T(A^{-1})^T\mathcal{I}_{t_0}^{t}(D)\rangle_\Omega)^{-1}A^T \\ &= \langle \mathcal{I}_{t_0}^{t+\Delta t}(D)\boldsymbol{F}\cdot\boldsymbol{F}^T\mathcal{I}_{t_0}^{t}(D)\rangle_\Omega(\langle \mathcal{I}_{t_0}^{t}(D)\boldsymbol{F}\cdot\boldsymbol{F}^T\mathcal{I}_{t_0}^{t}(D)\rangle_\Omega)^{-1} \\ &= e^{-D\Delta t}. \end{aligned} \tag{B38}$$

To obtain the last equality above, we used eq. (B37). The normalized correlation matrix is equal to the perturbation matrix and thus proves the FDT (see section 6 and appendixes C1 and C2)).

### B11 Experiments with stochastic forcing

### B12 Experiment L1W

For the stochastic forcing straightforward calculations, based on eqs. (A11), (B34) and (B35), and supposing that $t-t_0 \gg S^{-1}$, that is: $e^{-S(t-t_0)} \approx 0$, leads to the correlation matrix:

$$\begin{aligned} C(\Delta t) &= \begin{pmatrix} \langle u_a(t)u_a(t+\Delta t)\rangle_\Omega & \langle u_o(t)u_a(t+\Delta t)\rangle_\Omega \\ \langle u_a(t)u_o(t+\Delta t)\rangle_\Omega & \langle u_o(t)u_o(t+\Delta t)\rangle_\Omega \end{pmatrix} = \langle \begin{pmatrix} u_a(t+\Delta t) \\ u_o(t+\Delta t) \end{pmatrix} \cdot (u_a(t), u_o(t))\rangle_\Omega \\ &= \langle \begin{pmatrix} \mathcal{I}_{t_0}^{t+\Delta t}(Sm) & 0 \\ \frac{-\mathcal{I}_{t_0}^{t+\Delta t}(Sm)+\mathcal{I}_{t_0}^{t+\Delta t}(0)}{m} & \mathcal{I}_{t_0}^{t+\Delta t}(0) \end{pmatrix} \begin{pmatrix} 1 & 0 \\ 0 & 0 \end{pmatrix} \begin{pmatrix} \mathcal{I}_{t_0}^{t}(Sm) & \frac{-\mathcal{I}_{t_0}^{t}(Sm)+\mathcal{I}_{t_0}^{t}(0)}{m} \\ 0 & \mathcal{I}_{t_0}^{t}(0) \end{pmatrix} \rangle_\Omega \\ &= \langle \begin{pmatrix} \mathcal{I}_{t_0}^{t}(Sm)\mathcal{I}_{t_0}^{t+\Delta t}(Sm) & \mathcal{I}_{t_0}^{t+\Delta t}(Sm)\frac{-\mathcal{I}_{t_0}^{t}(Sm)+\mathcal{I}_{t_0}^{t}(0)}{m} \\ \mathcal{I}_{t_0}^{t}(Sm)\frac{-\mathcal{I}_{t_0}^{t+\Delta t}(Sm)+\mathcal{I}_{t_0}^{t+\Delta t}(0)}{m} & \frac{(-\mathcal{I}_{t_0}^{t+\Delta t}(Sm)+\mathcal{I}_{t_0}^{t+\Delta t}(0))(-\mathcal{I}_{t_0}^{t}(Sm)+\mathcal{I}_{t_0}^{t}(0))}{m^2} \end{pmatrix} \rangle_\Omega \\ &= R\begin{pmatrix} \frac{e^{-Sm\Delta t}}{Sm} & \frac{e^{-Sm\Delta t}}{Sm^2} \\ \frac{2e^{-Sm\Delta t}-1}{Sm^2} & -\frac{2+e^{-Sm\Delta t}}{Sm^3}+\frac{2(t-t_0)}{m^2} \end{pmatrix}. \end{aligned} \tag{B39}$$

Straight forward calculations show that $C(\Delta t)C(0)^{-1}$ is equal to the perturbation matrix (A12) which profs the FDT.

### B13 Experiment LW2

For the stochastic forcing straightforward calculations, based on eqs. (A18), (B34) and (B35), lead to the correlation matrix:

$$
\begin{aligned}
C(\Delta t) &= \begin{pmatrix} \langle u_a(t)u_a(t+\Delta t)\rangle_\Omega & \langle u_o(t)u_a(t+\Delta t)\rangle_\Omega \\ \langle u_a(t)u_o(t+\Delta t)\rangle_\Omega & \langle u_o(t)u_o(t+\Delta t)\rangle_\Omega \end{pmatrix} = \langle \begin{pmatrix} u_a(t+\Delta t) \\ u_o(t+\Delta t) \end{pmatrix} \cdot (u_a(t), u_o(t))\rangle_\Omega \\
&= \langle \begin{pmatrix} \mathcal{I}_{t_0}^{t+\Delta t}(Sm) & 0 \\ \frac{-\mathcal{I}_{t_0}^{t+\Delta t}(Sm)+\mathcal{I}_{t_0}^{t+\Delta t}(S)}{m-1} & \mathcal{I}_{t_0}^{t+\Delta t}(S) \end{pmatrix} \begin{pmatrix} 1 & 0 \\ 0 & 0 \end{pmatrix} \begin{pmatrix} \mathcal{I}_{t_0}^{t}(Sm) & \frac{-\mathcal{I}_{t_0}^{t}(Sm)+\mathcal{I}_{t_0}^{t}(S)}{m-1} \\ 0 & \mathcal{I}_{t_0}^{t}(S) \end{pmatrix}\rangle_\Omega \\
&= \langle \begin{pmatrix} \mathcal{I}_{t_0}^{t}(Sm)\mathcal{I}_{t_0}^{t+\Delta t}(Sm) & \mathcal{I}_{t_0}^{t+\Delta t}(Sm)\frac{-\mathcal{I}_{t_0}^{t}(Sm)+\mathcal{I}_{t_0}^{t}(S)}{m-1} \\ \mathcal{I}_{t_0}^{t}(Sm)\frac{-\mathcal{I}_{t_0}^{t+\Delta t}(Sm)+\mathcal{I}_{t_0}^{t+\Delta t}(S)}{m-1} & \frac{(-\mathcal{I}_{t_0}^{t+\Delta t}(Sm)+\mathcal{I}_{t_0}^{t+\Delta t}(S))(-\mathcal{I}_{t_0}^{t}(Sm)+\mathcal{I}_{t_0}^{t}(S))}{(m-1)^2} \end{pmatrix}\rangle_\Omega \\
&= R \begin{pmatrix} \frac{e^{-Sm\Delta t}}{Sm} & \frac{e^{-Sm\Delta t}}{SMm} \\ \frac{1}{S(m-1)}(\frac{2e^{-S\Delta t}}{M} - \frac{e^{-Sm\Delta t}}{m}) & \frac{1}{SM(m-1)}(e^{-S\Delta t} - \frac{e^{-Sm\Delta t}}{m}) \end{pmatrix}.
\end{aligned}
\tag{B40}
$$

Straight forward calculations show that $C(\Delta t)C(0)^{-1}$ is equal to the perturbation matrix (A19) which profs the FDT.

### B14 Experiment LW3

For the stochastic forcing straightforward calculations, based on eq. (A25), (B34) and (B35), lead to the correlation matrix:

$$
\begin{aligned}
C(\Delta t) &= \begin{pmatrix} \langle u_a(t)u_a(t+\Delta t)\rangle_\Omega & \langle u_o(t)u_a(t+\Delta t)\rangle_\Omega \\ \langle u_a(t)u_o(t+\Delta t)\rangle_\Omega & \langle u_o(t)u_o(t+\Delta t)\rangle_\Omega \end{pmatrix} = \langle \begin{pmatrix} u_a(t+\Delta t) \\ u_o(t+\Delta t) \end{pmatrix} \cdot (u_a(t), u_o(t))\rangle_\Omega \\
&= \langle \frac{1}{M} \begin{pmatrix} m\mathcal{I}_{t_0}^{t+\Delta t}(SM)+\mathcal{I}_{t_0}^{t+\Delta t}(0) & -m\mathcal{I}_{t_0}^{t+\Delta t}(SM)+m\mathcal{I}_{t_0}^{t+\Delta t}(0) \\ -\mathcal{I}_{t_0}^{t+\Delta t}(SM)+\mathcal{I}_{t_0}^{t+\Delta t}(0) & \mathcal{I}_{t_0}^{t+\Delta t}(SM)+m\mathcal{I}_{t_0}^{t+\Delta t}(0) \end{pmatrix} \begin{pmatrix} 1 & 0 \\ 0 & 0 \end{pmatrix} \\
&\qquad \frac{1}{M} \begin{pmatrix} m\mathcal{I}_{t_0}^{t}(SM)+\mathcal{I}_{t_0}^{t}(0) & -\mathcal{I}_{t_0}^{t}(SM)+\mathcal{I}_{t_0}^{t}(0) \\ -m\mathcal{I}_{t_0}^{t}(SM)+m\mathcal{I}_{t_0}^{t}(0) & \mathcal{I}_{t_0}^{t}(SM)+m\mathcal{I}_{t_0}^{t}(0) \end{pmatrix}\rangle_\Omega \\
&= \frac{e^{-SM\Delta t}}{M^2}\langle \begin{pmatrix} m\mathcal{I}_{t_0}^{t}(SM)(m\mathcal{I}_{t_0}^{t}(SM)+\mathcal{I}_{t_0}^{t}(0)) & m\mathcal{I}_{t_0}(SM)(-\mathcal{I}_{t_0}^{t}(SM)+\mathcal{I}_{t_0}^{t}(0)) \\ -\mathcal{I}_{t_0}(SM)(m\mathcal{I}_{t_0}^{t}(SM)+\mathcal{I}_{t_0}^{t}(0)) & -\mathcal{I}_{t_0}^{t}(SM)(-\mathcal{I}_{t_0}^{t}(SM)+\mathcal{I}_{t_0}^{t}(0)) \end{pmatrix}\rangle_\Omega \\
&\qquad + \frac{1}{M^2}\langle \begin{pmatrix} \mathcal{I}_{t_0}^{t}(0)(m\mathcal{I}_{t_0}^{t}(SM)+\mathcal{I}_{t_0}^{t}(0)) & \mathcal{I}_{t_0}^{t}(0)(-\mathcal{I}_{t_0}^{t}(SM)+\mathcal{I}_{t_0}^{t}(0)) \\ \mathcal{I}_{t_0}(0)(m\mathcal{I}_{t_0}^{t}(SM)+\mathcal{I}_{t_0}^{t}(0)) & \mathcal{I}_{t_0}^{t}(0)(-\mathcal{I}_{t_0}^{t}(SM)+\mathcal{I}_{t_0}^{t}(0)) \end{pmatrix}\rangle_\Omega \\
&= \frac{R}{M^2}(\frac{e^{-SM\Delta t}}{SM} \begin{pmatrix} m(m+2) & m \\ -m-2 & -1 \end{pmatrix} + \frac{2}{SM} \begin{pmatrix} m & -1 \\ m & -1 \end{pmatrix} + 2(t-t_0) \begin{pmatrix} 1 & 1 \\ 1 & 1 \end{pmatrix}).
\end{aligned}
\tag{B41}
$$

Straight forward calculations show that $C(\Delta t)C(0)^{-1}$ is equal to the perturbation matrix (A26) which profs the FDT.

### B15 Experiment L1C

$$\begin{aligned} \partial_t \tilde{F} &= - \mu \tilde{F} + F && \text{(B42)} \\ \partial_t u_a &= - Sm u_a + \tilde{F} && \text{(B43)} \\ \partial_t u_o &= S \; u_a && \text{(B44)} \end{aligned}$$

$$\partial_t \begin{pmatrix} \tilde{F} \\ u_a \\ u_o \end{pmatrix} = \begin{pmatrix} -\mu & 0 & 0 \\ 1 & -Sm & 0 \\ 0 & S & 0 \end{pmatrix} \begin{pmatrix} \tilde{F} \\ u_a \\ u_o \end{pmatrix} + \begin{pmatrix} F \\ 0 \\ 0 \end{pmatrix} \tag{B45}$$

$$\lambda_1 = -\mu, \; e_1 = \begin{pmatrix} \mu(\mu - Sm) \\ -\mu \\ S \end{pmatrix}; \; \lambda_2 = -Sm, \; e_2 = \begin{pmatrix} 0 \\ m \\ -1 \end{pmatrix}; \; \lambda_3 = 0, \; e_3 = \begin{pmatrix} 0 \\ 0 \\ 1 \end{pmatrix} \tag{B46}$$

$$P = ADA^{-1} = \begin{pmatrix} -\mu & 0 & 0 \\ 1 & -Sm & 0 \\ 0 & S & 0 \end{pmatrix} =$$

$$\begin{pmatrix} \mu(\mu - Sm) & 0 & 0 \\ -\mu & m & 0 \\ S & -1 & 1 \end{pmatrix} \begin{pmatrix} -\mu & 0 & 0 \\ 0 & -Sm & 0 \\ 0 & 0 & 0 \end{pmatrix} \begin{pmatrix} [\mu(\mu - Sm)]^{-1} & 0 & 0 \\ [m(\mu - Sm)]^{-1} & m^{-1} & 0 \\ [\mu m]^{-1} & m^{-1} & 1 \end{pmatrix}. \tag{B47}$$

The solution is:

$$\begin{pmatrix} \tilde{F} \\ u_a(t) \\ u_o(t) \end{pmatrix} = A\mathcal{I}_{t_0}^t(D)A^{-1} \begin{pmatrix} 1 \\ 0 \\ 0 \end{pmatrix} dt'$$

$$= A \begin{pmatrix} \mathcal{I}_{t_0}^t(\mu) & 0 & 0 \\ 0 & \mathcal{I}_{t_0}^t(Sm) & 0 \\ 0 & 0 & \mathcal{I}_{t_0}^t(0) \end{pmatrix} A^{-1} \begin{pmatrix} 1 \\ 0 \\ 0 \end{pmatrix}$$

$$= \begin{pmatrix} \mathcal{I}_{t_0}^t(\mu) \\ \left[-\mathcal{I}_{t_0}^t(\mu) + \mathcal{I}_{t_0}^t(Sm)\right]/(\mu - Sm) \\ \left[Sm\mathcal{I}_{t_0}^t(\mu) - \mu\mathcal{I}_{t_0}^t(Sm) + (\mu - Sm)\mathcal{I}_{t_0}^t(0)\right]/[m\mu(\mu - Sm)] \end{pmatrix}. \tag{B48}$$

From this follows (after dropping decaying exponentials):

$$\langle u_a^2(t)\rangle_\Omega = \frac{R}{Sm\mu(\mu+Sm)} \tag{B49}$$

$$\langle u_a(t)\tilde{F}(t)\rangle_\Omega = \frac{R}{\mu(\mu+Sm)} \tag{B50}$$

$$\langle u_o^2(t)\rangle_\Omega = \frac{R(3Sm^3+2Sm^2\mu-2Sm\mu^2-3\mu^3)}{\mu^3m^3(\mu^2-Sm^2)} + \frac{2R(t-t_0)}{\mu^2m^2} \tag{B51}$$

$$\langle u_a(t)u_o(t)\rangle_\Omega = \frac{R}{S\mu^2m^2} \tag{B52}$$

In the absence of forcing an initial perturbation at $t=0$, $(\bar{\tilde{F}},\bar{u}_a,\bar{u}_o)$ evolves as:

$$\begin{pmatrix}\tilde{F}^P(t)\\ u_a^P(t)\\ u_o^P(t)\end{pmatrix} = \begin{pmatrix} e^{-\mu t} & 0 & 0\\ \frac{e^{-Smt}-e^{-\mu t}}{\mu-Sm} & e^{-Smt} & 0\\ \frac{\mu(1-e^{-Smt})-Sm(1-e^{-\mu t})}{\mu m(\mu-Sm)} & \frac{1-e^{-Smt}}{m} & 1\end{pmatrix}\begin{pmatrix}\bar{\tilde{F}}\\ \bar{u}_a\\ \bar{u}_o\end{pmatrix}. \tag{B53}$$

The perturbation matrix above is obtained by calculating $A\exp(Dt)A^{-1}$. Note that the lower-right $2\times 2$ sub-matrix is identical to eq. (A12), a simple consequence of linearity.

## B16 Experiment L2C

$$\partial_t\tilde{F} = -\mu\tilde{F}+F \tag{B54}$$

$$\partial_t u_a = -Smu_a+\tilde{F} \tag{B55}$$

$$\partial_t u_o = S\,(u_a-u_o) \tag{B56}$$

$$\partial_t\begin{pmatrix}\tilde{F}\\ u_a\\ u_o\end{pmatrix} = \begin{pmatrix}-\mu & 0 & 0\\ 1 & -Sm & 0\\ 0 & S & -S\end{pmatrix}\begin{pmatrix}\tilde{F}\\ u_a\\ u_o\end{pmatrix} + \begin{pmatrix}F\\ 0\\ 0\end{pmatrix} \tag{B57}$$

$$\lambda_1 = -\mu,\ e_1 = \begin{pmatrix}(Sm-\mu)(S-\mu)\\ (S-\mu)\\ S\end{pmatrix};\ \lambda_2 = -Sm,\ e_2 = \begin{pmatrix}0\\ m-1\\ -1\end{pmatrix};\ \lambda_3 = -S,\ e_3 = \begin{pmatrix}0\\ 0\\ 1\end{pmatrix} \tag{B58}$$

$$P = ADA^{-1} = \begin{pmatrix} -\mu & 0 & 0 \\ 1 & -Sm & 0 \\ 0 & S & -S \end{pmatrix} =$$
$$\begin{pmatrix} (Sm-\mu)(S-\mu) & 0 & 0 \\ (S-\mu) & m-1 & 0 \\ S & -1 & 1 \end{pmatrix} \begin{pmatrix} -\mu & 0 & 0 \\ 0 & -Sm & 0 \\ 0 & 0 & -S \end{pmatrix}$$
$$\begin{pmatrix} [(Sm-\mu)(S-\mu)]^{-1} & 0 & 0 \\ -[(m-1)(Sm-\mu)]^{-1} & (m-1)^{-1} & 0 \\ -[(m-1)(S-\mu)]^{-1} & (m-1)^{-1} & 1 \end{pmatrix}. \tag{B59}$$

The solution is:

$$\begin{pmatrix} \tilde{F} \\ u_a(t) \\ u_o(t) \end{pmatrix} = A\mathcal{I}_{t_0}^t(D)A^{-1}\begin{pmatrix} 1 \\ 0 \\ 0 \end{pmatrix}$$
$$= A\begin{pmatrix} \mathcal{I}_{t_0}^t(\mu) & 0 & 0 \\ 0 & \mathcal{I}_{t_0}^t(Sm) & 0 \\ 0 & 0 & \mathcal{I}_{t_0}^t(S) \end{pmatrix} A^{-1}\begin{pmatrix} 1 \\ 0 \\ 0 \end{pmatrix}$$
$$= \begin{pmatrix} \mathcal{I}_{t_0}^t(\mu) \\ \left[\mathcal{I}_{t_0}^t(\mu) - \mathcal{I}_{t_0}^t(Sm)\right]/(Sm-\mu) \\ \left[S(m-1)\mathcal{I}_{t_0}^t(\mu) + (S-\mu)\mathcal{I}_{t_0}^t(Sm) - (Sm-\mu)\mathcal{I}_{t_0}^t(S)\right]/[(m-1)(Sm-\mu)(S-\mu)] \end{pmatrix}. \tag{B60}$$

From this follows (after dropping decaying exponentials):

$$\langle u_a^2(t)\rangle_\Omega = \frac{R}{\mu Sm(Sm+\mu)} \tag{B61}$$
$$\langle u_a(t)\tilde{F}(t)\rangle_\Omega = \frac{R}{\mu(Sm+\mu)} \tag{B62}$$
$$\langle u_o^2(t)\rangle_\Omega = \langle u_a(t)u_o(t) = \frac{R(SM+\mu)}{mSM\mu(S+\mu)(Sm+\mu)}. \tag{B63}$$

In the absence of forcing an initial perturbation at $t=0$, $(\bar{\tilde{F}}, \bar{u}_a, \bar{u}_o)$ evolves as:

$$\begin{pmatrix} \tilde{F}^P(t) \\ u_a^P(t) \\ u_o^P(t) \end{pmatrix} = \begin{pmatrix} e^{-\mu t} & 0 & 0 \\ \frac{e^{-Smt}-e^{-\mu t}}{\mu - Sm} & e^{-Smt} & 0 \\ \frac{(m-1)e^{-\mu t}+(S-\mu)e^{-Smt}-(Sm-\mu)e^{-St}}{(m-1)(S-\mu)(Sm-\mu)} & \frac{e^{-St}-e^{-Smt}}{m-1} & e^{-St} \end{pmatrix} \begin{pmatrix} \bar{\tilde{F}} \\ \bar{u}_a \\ \bar{u}_o \end{pmatrix}. \tag{B64}$$

The perturbation matrix above is obtained by calculating $A\exp(Dt)A^{-1}$. Note that the lower-right $2\times 2$ sub-matrix is identical to eq. (A19), a simple consequence of linearity.

### B17 Experiment L3C

Full interaction both ways.

$$\partial_t \tilde{F} = -\mu\tilde{F} + F \tag{B65}$$

$$\partial_t u_a = -Sm(u_a - u_o) + \tilde{F} \tag{B66}$$

$$\partial_t u_o = S(u_a - u_o) \tag{B67}$$

$$\partial_t \begin{pmatrix} \tilde{F} \\ u_a \\ u_o \end{pmatrix} = \begin{pmatrix} -\mu & 0 & 0 \\ 1 & -Sm & Sm \\ 0 & S & -S \end{pmatrix} \begin{pmatrix} \tilde{F} \\ u_a \\ u_o \end{pmatrix} + \begin{pmatrix} F \\ 0 \\ 0 \end{pmatrix} \tag{B68}$$

$$\lambda_1 = -\mu,\ e_1 = \begin{pmatrix} \mu(\mu - SM) \\ S-\mu \\ S \end{pmatrix};\ \lambda_2 = -SM,\ e_2 = \begin{pmatrix} 0 \\ m \\ -1 \end{pmatrix};\ \lambda_3 = 0,\ e_3 = \begin{pmatrix} 0 \\ 1 \\ 1 \end{pmatrix} \tag{B69}$$

$$P = ADA^{-1} = \begin{pmatrix} -\mu & 0 & 0 \\ 1 & -Sm & Sm \\ 0 & S & -S \end{pmatrix} =$$

$$\begin{pmatrix} \mu(\mu-SM) & 0 & 0 \\ S-\mu & m & 1 \\ S & -1 & 1 \end{pmatrix} \begin{pmatrix} -\mu & 0 & 0 \\ 0 & -SM & 0 \\ 0 & 0 & 0 \end{pmatrix} \begin{pmatrix} [\mu(\mu-SM)]^{-1} & 0 & 0 \\ [M(\mu-SM)]^{-1} & M^{-1} & -M^{-1} \\ (\mu M)^{-1} & M^{-1} & mM^{-1} \end{pmatrix}. \tag{B70}$$

The solution is:

$$\begin{pmatrix} \tilde{F} \\ u_a(t) \\ u_o(t) \end{pmatrix} = A\mathcal{I}_{t_0}^t(D)A^{-1} \begin{pmatrix} 1 \\ 0 \\ 0 \end{pmatrix} dt'$$

$$= A \begin{pmatrix} \mathcal{I}_{t_0}^t(\mu) & 0 & 0 \\ 0 & \mathcal{I}_{t_0}^t(SM) & 0 \\ 0 & 0 & \mathcal{I}_{t_0}^t(0)dt' \end{pmatrix} A^{-1} \begin{pmatrix} 1 \\ 0 \\ 0 \end{pmatrix}$$

$$= \begin{pmatrix} \mathcal{I}_{t_0}^t(\mu) \\ \left[M(S-\mu)\mathcal{I}_{t_0}^t(\mu) + m\mu\mathcal{I}_{t_0}^t(SM) + (\mu-SM)\mathcal{I}_{t_0}^t(0)\right]/[M\mu(\mu-SM)] \\ \left[SM\mathcal{I}_{t_0}^t(\mu) - \mu\mathcal{I}_{t_0}^t(SM) + (\mu-SM)\mathcal{I}_{t_0}^t(0)\right]/[M\mu(\mu-SM)] \end{pmatrix}. \tag{B71}$$

From this follows (after dropping decaying exponentials):

$$\langle u_a^2(t)\rangle_\Omega = \frac{R(-3S^2M^2+(2m^2-m-3)S\mu+m(m+4)\mu^2)}{M^3S\mu^3(SM+\mu)}+\frac{2R(t-t_0)}{\mu^2M^2} \quad \text{(B72)}$$

$$\langle u_a(t)\tilde{F}(t)\rangle_\Omega = \frac{R(\mu+S)}{\mu^2(SM+\mu)} \quad \text{(B73)}$$

$$\langle u_o^2(t)\rangle_\Omega = \frac{-R(3S^2M^2+5(m+1)S\mu+3\mu^2)}{M^3S\mu^3(SM+\mu)}+\frac{2R(t-t_0)}{\mu^2M^2} \quad \text{(B74)}$$

$$\langle u_a(t)u_o(t)\rangle_\Omega = \frac{R(-3S^2M^2+(m^2-3m-4)S\mu+(m-2)\mu^2)}{M^3S\mu^3(SM+\mu)}+\frac{2R(t-t_0)}{\mu^2M^2} \quad \text{(B75)}$$

In the absence of forcing an initial perturbation at $t=0$, $(\bar{\tilde{F}},\bar{u}_a,\bar{u}_o)$ evolves as:

$$\begin{pmatrix}\tilde{F}^P(t)\\ u_a^P(t)\\ u_o^P(t)\end{pmatrix} = \begin{pmatrix} e^{-\mu t} & 0 & 0\\ \frac{M(S-\mu)e^{-\mu t}+m\mu e^{-SMt}+(\mu-SM)}{\mu M(\mu-SM)} & \frac{1+me^{-SMt}}{M} & \frac{m(1-e^{-SMt})}{M}\\ \frac{SMe^{-\mu t}-\mu e^{-SMt}-(SM-\mu)}{\mu M(\mu-SM)} & \frac{1-e^{-SMt}}{M} & \frac{e^{-SMt}+m}{M}\end{pmatrix}\begin{pmatrix}\bar{\tilde{F}}\\ \bar{u}_a\\ \bar{u}_o\end{pmatrix}. \quad \text{(B76)}$$

The perturbation matrix above is obtained by calculating $A\exp(Dt)A^{-1}$. Note that the lower-right $2\times2$ sub-matrix is identical to eq. (A26), a simple consequence of linearity.

## Appendix C: Fluctuation Dissipation Theorem

The fluctuation dissipation theorem applies to a system, if the system relaxes from a forced state to the unforced dynamics, in the same manner as if the forced state were due to an internal fluctuation of the system.

The average response of a system to an external small amplitude forcing is:

$$\langle v(t,u)\rangle_\Omega = \langle u(t)\rangle_\Omega + \int_{-\infty}^{t}\tilde{\mu}(t,s)F(s)ds + O(F^2), \quad \text{(C1)}$$

where the first term on the r.h.s. is the unforce dynamics. The upper bound of the integral is imposed by causality. In the linear case only the first term in the Taylor expansion of the perturbation has to be considered ($O(F^2)=0$) and we can put $\langle u(t)\rangle_\Omega = 0$ and $v(t,u)=v(t)$ as the evolution does not depend on the state $u$. When the system is stationary we can simplify to $\mu(t-s)=\tilde{\mu}(t,s)$.

### C1 Example: Langevin equation (white noise)

The Ornstein-Uhlenbeck process ($S>0$) and Brownian motion ($S=0$) is considered:

$$\partial_t u_O = -Su_O + F. \quad \text{(C2)}$$

The response function is, using eq. (A3):

$$\mu(\Delta t) = \exp(-S\Delta t). \quad \text{(C3)}$$

Straight forward calculations using eq. (B27) show that an initial perturbation decreases as:

$$\frac{u_O(t_0+\Delta t)}{u_O(t_0)} = \chi(\Delta t) = \exp(-S\Delta t). \tag{C4}$$

On the other hand we can show, using the same equation, that time correlation is:

$$C(t_0,\Delta t) = \langle u_O(t_0) u_O(t_0+\Delta t)\rangle_\Omega = \exp(-S\Delta t) \tag{C5}$$

and the normalized correlation matrix is:

$$\tilde{C}(\Delta t) = C(t_0,\Delta t) C(t_0,0)^{-1} = \exp(-S\Delta t) \tag{C6}$$

This leads to:

$$\mu(\Delta t) = \tilde{C}(\Delta t) = \chi(\Delta t). \tag{C7}$$

Historically the first equality is the first FDT and the second equality is Onsager's principle (see Barrat and Hansen (2003)). Today the second equality which matches the decay of an initial perturbation with the normalized correlation matrix is referred to as the FDT.

Note that for Brownian motion ($S = 0$) perturbations do not decay and the process has the martingale property, but also in this case the normalized correlation matrix does not depend on the absolute time $t_0$ as the process is of stationary increment.

### C2 Example: Langevin equation with colored noise

$$\begin{aligned} \partial_t \tilde{F} &= - \mu \tilde{F} + F && \text{(C8)} \\ \partial_t u_{OC} &= - S u_{OC} + \tilde{F}. && \text{(C9)} \end{aligned}$$

The mathematical structure is the same as the L2 model and the solution is:

$$\tilde{F}(t) = \int_{t_0}^{t} e^{\mu(t'-t)} F(t') dt' + e^{\mu(t_0-t)} \tilde{F}(t_0) \tag{C10}$$

$$\begin{aligned} u_{OC}(t) &= \int_{t_0}^{t} e^{S(t'-t)} \tilde{F}(t') dt' + e^{S(t_0-t)} u_{OC}(t_0) \\ &= \frac{1}{\mu - S} \int_{t_0}^{t} (e^{S(t'-t)} - e^{\mu(t'-t)}) F(t') dt' \\ &+ e^{S(t_0-t)} u_{OC}(t_0) + \frac{\tilde{F}(t_0)}{\mu - S} (e^{S(t_0-t)} - e^{\mu(t_0-t)}). \end{aligned} \tag{C11}$$

It is important to see that the system composed of eqs. (C8) and (C9) is forced by a white noise and does obey the FDT (in 2D space), when only eq. (C9) is considered forced by a colored noise the FDT (in 1D space) is not observed.

Open Access EGU

More precisely, in 2D space: a perturbation $(\bar{F}, \bar{u}_{OC})$ (putting $F = 0$ in eqs. (C10) and (C11)) decreases as:

$$\begin{pmatrix} \tilde{F}^P(\Delta t) \\ u_{OC}^P(\Delta t) \end{pmatrix} = \chi(\Delta t) \begin{pmatrix} \bar{F} \\ \bar{u}_{OC} \end{pmatrix} = \begin{pmatrix} e^{-\mu\Delta t} & 0 \\ \frac{e^{-\mu\Delta t} - e^{-S\Delta t}}{S-\mu} & e^{-S\Delta t} \end{pmatrix} \begin{pmatrix} \bar{F} \\ \bar{u}_{OC} \end{pmatrix}. \tag{C12}$$

Eq. (C11) shows that: $\mu(\Delta t) = \chi(\Delta t)$.

The time-lagged correlation matrix is:

$$\begin{aligned} C(t) &= \begin{pmatrix} \langle \tilde{F}(t_0)\tilde{F}(t_0+\Delta t)\rangle_\Omega & \langle u_{OC}(t_0)\tilde{F}(t_0+\Delta t)\rangle_\Omega \\ \langle \tilde{F}(t_0)u_{OC}(t_0+\Delta t)\rangle_\Omega & \langle u_{OC}(t_0)u_{OC}(t_0+\Delta t)\rangle_\Omega \end{pmatrix} \\ &= R \begin{pmatrix} \frac{e^{-\mu\Delta t}}{\mu} & \frac{e^{-\mu\Delta t}}{\mu(S+\mu)} \\ 2\frac{e^{-\mu\Delta t} - e^{-S\Delta t}}{S^2-\mu^2} + \frac{e^{-\mu\Delta t}}{\mu(S+\mu)} & \frac{Se^{-\mu\Delta t} - \mu e^{-S\Delta t}}{\mu S(S^2-\mu^2)} \end{pmatrix}. \end{aligned} \tag{C13}$$

Note that: $\partial_t \langle u_{OC}(t_0)u_{OC}(t_0+t)\rangle_\Omega|_{t=0} = 0$, so that contrary to the white-noise case the correlation is differentiable at $t = 0$.

Calculations give:

$$C(0)^{-1} = R^{-1} \begin{pmatrix} S+\mu & -S(S+\mu) \\ -S(S+\mu) & S(S+\mu)^2 \end{pmatrix} \tag{C14}$$

and the normalized correlation matrix:

$$\tilde{C}(\Delta t) = C(\Delta t)C(0)^{-1} = \begin{pmatrix} e^{-\mu\Delta t} & 0 \\ \frac{e^{-\mu\Delta t} - e^{-S\Delta t}}{S-\mu} & e^{-S\Delta t} \end{pmatrix}. \tag{C15}$$

As for the white noise case we get:

$$\mu(\Delta t) = \tilde{C}(\Delta t) = \chi(\Delta t). \tag{C16}$$

The first equality is the (first) FDT and the second equality is Onsager's principle.

When only eq. (C9) forced by a colored noise is considered the FDT does not apply. Indeed, an imposed perturbation $\bar{u}_{OC}$ still has the same decay of the white noise case given by eq. (C4), as the decay in the linear equation does not depend on the noise. The response function is also identical to eq. (C3), as the response in the linear equation does not depend on the noise. It follows, that $\mu_u(\Delta t) = \chi_u(\Delta t)$.

Whereas the scalar calculations give:

$$\tilde{C}_u(\Delta t) = \frac{\langle u_{OC}(t_0)u_{OC}(t_0+\Delta t)\rangle_\Omega}{\langle u_{OC}^2(t_0)\rangle_\Omega} = \frac{Se^{-\mu\Delta t} - \mu e^{-S\Delta t}}{S-\mu}, \tag{C17}$$

which does not agree with the response function or the decay law of a perturbation and so neither the FDT nor Onsager's principle is observed. The failure of the FDT is due to the non-vanishing auto-correlation time of $\tilde{F}$. A consequence of this is that $\langle u_{OC}(t_0)\tilde{F}(t_0+t))\rangle_\Omega \neq 0$ even if $t > 0$, meaning that the future forcing is correlated to the actual state.

The FDT applies only when the forcing correlation time vanishes, that is $\mu \gg S$ or a generalized Langevin equation is used, that is the friction term is presented by a memory kernel and eq. (C9) replaced by:

$$\partial_t u_{OC}(t) = - \frac{S}{\mu} \int_{-\infty}^{t} e^{\mu(t'-t)} u_{OC}(t')dt' + \tilde{F}. \tag{C18}$$

A more general discussion of the problem of causality due to time-correlated noise and the generalized Langevin equation are given by Balakrishnan (1979).

As the system is linear the pdf's of the variables are Gaussian. In the unperturbed system averages vanish and second-order moments are given in eq. (C13) by setting $t = 0$.

*Author contributions.* AW has performed the coding, the research and the writing of the paper

*Competing interests.* AW has no competing interest

*Acknowledgements.* This work was funded by Labex OASUG@2020 (Investissement d'avenir - ANR10 LABX56).

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
