# Peer review of "On fluctuating air-sea-interaction in local models: linear theory"

_Geoscientific Model Development, 2018_

## Referee Comment (RC1) · Anonymous Referee #1 · 18 Mar 2019

This study uses a set of idealized linear models of air-sea interaction to assess the importance of including ocean velocity in computations of surface momentum fluxes. Fluctuation-dissipation results are also discussed for these models.

I am unable to recommend publication of this study in its present form. The fact that accounting for surface currents (or not doing so) changes the surface momentum budget is true by definition; the existence of consequences for the energetics is also well-known. The models used here are not needed to demonstrate these points, and are too simple to quantify the implications meaningfully. In particular, the models' assumption that variations in surface currents are entirely driven by local momentum fluxes from the atmosphere is far too large an oversimplification to allow for a meaningful quantification of the importance of these currents on surface coupling.

[Figure]

There is also nothing new in the fact that these linear models, when cast as linear stochastic differential equations with solutions which are Markov processes, satisfy fluctuation-dissipation theorems: they are after all just multivariate Ornstein-Uhlenbeck processes. The fact that non-Markov state variable subspaces (such as obtained in the case of red noise forcing without augmenting the state space appropriately) do not satisfy a FDT is also a standard result.

---

## Referee Comment (RC2) · Anonymous Referee #2 · 20 Mar 2019

General comments:

This manuscript examines the dynamics of three possible parameterizations of air-sea momentum transfer by treating them as stand-alone linear models that are either deterministic or driven by additive noise. For reasons listed below, I cannot recommend publication of this manuscript.

The author introduces three air-sea interaction models that appear much simpler than I suspect is actually implemented in coupled models. Citations to coupled models that use these parameterizations might have obviated this objection, but without them, I cannot say that detailed examination of the three cases as done here merits publication. The simplicity of these models does not warrant such detailed exposition in a published article, and the appendices are generally reproduceable by anyone who has

studied undergraduate linear algebra.

In spite of the general title, only momentum fluxes at the air-sea interface are considered. Clearly, thermal fluxes are just as important, and these are ignored completely. This manuscript treats only a small part of "air-sea-interaction in local models," in spite of the more general title.

Next, the stochastic models involve only additive random forcing. At the timescales at which these parameterizations should be used, multiplicative noise is likely to be much more appropriate. The analytical treatment of both additive and multiplicative stochastic linear models, some of which are inhomogeneous, have been treated in the geophysics literature for the last several decades. As it is not the reviewer's place to perform major revisions to a manuscript, I invite the author to perform a literature search for the easily-found references to such studies.

The same lack of citations applies to discussions of stochastic integration in both the Ito and Stratonovich sense, both of which have been used in weather/climate studies for years, if not decades. The fluctuation theorem has been used less often, but is still found in the mainstream weather and climate literature.

The author should be aware that he is not introducing the stochastic dynamics literature to geoscience. It is true that no one can read everything. However, this manuscript gives little or no citation either to the models whose parameterizations are featured here, nor to the vast amount of stochastic geophysical research that has already been published. Scholarship requires more.

Some specific comments:

Pg. 2, lines 14-15: A forced, dissipative system need not be confined to a strange attractor. Fixed points and limit cycles, as well as dense tori in a sampled phase space, are well-known in nature.

Pg. 4, Eq. (1) and beyond: Some symbols are never defined.

[Figure]

Pg. 6, lines 8-12: The author confuses momentum with inertia.

Pg. 6, line 12: Why is the additional timescale spurious? Most linear systems in nature do consist of a superposition of timescales.

Pg. 7 and beyond, including appendices: The author uses the term "solution" when what is meant is the "particular solution." Transients are part of the solution.

Most specific comments on the remainder of the manuscript are simply special cases of general comments stated above.
* * *

---

## Author Comment (AC1) · 20 Mar 2019

The main criticism of the reviewer is that the models are too simple and there is nothing new in my work. He performs this statement without citing a single reference showing that the presented results have been published elsewhere. This an expression of his personal view but not a scientific argumentation. The models I consider are also implemented as (local) "bulk formulas" in today's models of ocean and climate dynamics. The analytic calculations I present have, to the best of my knowledge not been published previously.

To increase scientific understanding of complex natural phenomena a hierarchy of models has to be constructed. When observational data and more involved models

are systematically connected to simpler models, which can be solved analytically, in such hierarchy, scientific understanding is increased. The models I discuss are highly idealised and can be (partly) solved analytically. They present a solid basis to which results from more involved models can be compared. There are many papers on the subject which analyse data from more or less idealised simulations, but very few (no?) analytic results. It is important, to my understanding to show how far analytic calculations can go. My work contains a large amount of analytical results (mostly in the appendices) and the reviewer has found no error in the calculations.

The reviewer states: "The fact that accounting for surface currents (or not doing so) changes the surface momentum budget is true by definition; the existence of consequences for the energetics is also wellknown" Yes, there are publications from observations and more involved models, but I give analytic results for the idealised models (bulk formulas), so one clearly sees where the differences come from. I am not aware that this has been done elsewhere. I also show that differences are qualitative not only quantitative, meaning that one parameterization can not mimic the behaviour of another parameterization by adapting eddy coefficients, which is attempted frequently. This has not been shown elsewhere.

The reviewer continues: "The models used here are not needed to demonstrate these points, and are too simple to quantify the implications meaningfully. In particular, the models' assump- tion that variations in surface currents are entirely driven by local momentum fluxes from the atmosphere is far too large an oversimplification to allow for a meaningful quantification of the importance of these currents on surface coupling." Yes, the models (bulk formulas) are local, but the scientific method is to split-up a complicated problem and to see where differences arise. It is important to distinguish between local and non-local contributions. To increase understanding both local and non-local contributions have to be studied separately. In my previous paper in JPO (https://doi.org/10.1175/JPO-D-17-0097.1), I compare data from one of the models to a non-local simulations and discuss the differences. And again: the models I consider

are implemented as (local) "bulk formulas" in toady's models of ocean and climate dynamics and of course non-local dynamics in the models non-linearly superpose to these local bulk formulas.

The reviewer writes: "There is also nothing new in the fact that these linear models, when cast as linear stochastic differential equations with solutions which are Markov processes, satisfy fluctuation-dissipation theorems: they are after all just multivariate Ornstein-Uhlenbeck processes. " The important point of my paper is that the fluctuation-dissipation theorems differ between the models, the relaxation time vary between different finite values and infinity, as for some models the solution includes a Brownian motion (which is not stationary like an Ornstein-Uhlenbeck processes). An infinite relaxation time points to long-time memory in the problem and questions ergodicity. This has not been stated elsewhere to the best of my knowledge. In my paper I give all the analytic results for the different bulk parameterizations, this has never been accomplished.

The reviewer further states that: "The fact that non-Markov state variable subspaces (such as obtained in the case of red noise forcing without augmenting the state space appropriately) do not satisfy a FDT is also a standard result." I agree and it is stated in my paper (last sentence of section 2): "Augmenting the phase space dimension to render a non-Markovian process Markovian is a standard procedure." (The statement of the reviewer is actually not correct. When a red-noise is used and the state space not-augmented the FDT can be assured by a generalised Langevin equation with a memory kernel, this point is discussed, e.g. in detail by Balakrishnan (1979), cited in my paper)

The reviewer does not say anything about the fluctuation-dissipation-relations that have been derived for the different models and compared. And he does not comment on the results concerning fluctuation theorems, which to the best of my knowledge, have never been considered in the context of environmental sciences.

The reviewer has taken a few aspects of a few of my results and qualified them as "not new" and rejects my paper on this grounds. He did not value the analytical results presented in my paper. I challenge the reviewer by giving me a single reference (other than my publication in JPO (https://doi.org/10.1175/JPO-D-17-0097.1) that discusses the fluctuation-dissipation relation OR the fluctuation-dissipation theorem OR fluctuation relations in the context of momentum transfer at the air-sea-interface, which are all discussed in the present work.

When presenting my previous results at conferences, the comment that arouse the most often was, on how these results change in different bulk formulas of air-sea interaction (as after my talk at the EGU2018 General Assembly). I performed this work to answer this question and submitted it to GMD so it reaches the right community.

---

## Author Comment (AC3) · 21 Mar 2019

The main criticism of the reviewer is that the models are too simple and the stochastic differential equations are extensively use in the community.

In climate sciences, there is a large body of work that adds noise to models of different complexity and analyses the results. But the point I want to make is that stochastic differential equations can actually be used to perform analytic calculations and obtain solid results. Therefore the appendices are detailed (also to allow for an easy verification of the results and show that part of it is indeed linear algebra). I am not aware that this has been done for air-sea interaction. Simple analytical models are wide-spread in the physics community but to the best of my knowledge, not for air-sea interaction

so far. When the reviewer talks about "The fluctuation theorem [...] is still found in the mainstream weather and climate literature", it is not clear to me if he means the fluctuation dissipation relation (section 4 of my paper), fluctuation dissipation theorem (section 6 of my paper) or fluctuation theorems (section 7 of my paper), which are all different. Explaining this differences and illustrating them is one purpose of my work. I challenge the reviewer to give me a single reference where fluctuation theorems are discussed for weather, ocean or climate.

I want to emphasise that the reviewer has found no error and given no reference showing that the results are not new.